

# On the Relationship between Teleconnections and Taiwan's Streamflow: Evidence of Climate Regime Shift and Implications for Seasonal Forecasting

Chia-Jeng Chen[1] and Tsung-Yu Lee[2]

[1]National Chung Hsing University, 145 Xingda Road, Taichung, Taichung 40227, Taiwan
[2]National Taiwan Normal University, 162 Heping East Road, Section 1, Taipei 10610, Taiwan
*Correspondence to:* Tsung-Yu Lee (tylee@ntnu.edu.tw)

**Abstract.** Interannual variations of catchment streamflow represent an integrated response to anomalies in regional moisture transport and atmospheric circulations, ultimately linked to large-scale climate oscillations. This study investigates the relationship between Taiwan's long-term summertime (July to September, JAS) streamflow and manifold teleconnection patterns. Lagged correlation analysis is conducted to calculate how JAS streamflow data derived at 28 upstream and 13 downstream gauges in Taiwan correlate with 14 teleconnection indices in the concurrent or preceding seasons. Out of the many indices, the West-Pacific and Pacific-Japan (PJ) patterns, both of which play a critical role in determining cyclonic activity in the western North Pacific basin, exhibit the highest concurrent correlations (most significant $r = 0.48$) with the JAS flows in Taiwan. At a one-month lead time, on the other hand, the Quasi-Biennial Oscillation significantly correlate with the JAS flows (most significant $r = -0.66$), indicating some forecasting utility. By further examining the correlation results using a 20-year moving window, peculiar temporal variations and possible climate regime shifts (CRS) can be revealed. To identify suspicious, abrupt changes in the correlation, a CRS test is employed. The late 1970s and 1990s are identified as two significant change points, and during the intermediate period, a marked in-phase relationship ($r \simeq 0.9$) between Taiwan's streamflow and the PJ index is observed. It is verified that the two shifts are in concordance with the alteration of large-scale circulations in the Pacific basin. Discussion about the changes in pattern correlation and composite maps before and after the change point is carried out, and our results suggest that empirical forecasting techniques should take into account the effect of CRS on predictor screening.

## 1 Introduction

Hydro-climatic forecasting is a crucial issue, particularly for those regions suffering increased rainfall intensity and/or reoccurring, persistent droughts under the condition of global warming (e.g., Liu et al., 2009). An emerging method for hydro-climatic forecasting is the usage of "teleconnection" patterns (van den Dool, 2007) that signify the influence of low-frequency climate oscillations on hydro-climates in remote locations by emanating shifts in meteorological systems (e.g., planetary waves, jet streams, and monsoons). Prominent teleconnection patterns have been proven useful for the prediction of regional climates with lead times from weeks to months (e.g., Palmer and Anderson, 1994; Chiew et al., 1998; Goddard et al., 2001). In the equatorial Pacific, El Niño-Southern Oscillation (ENSO) stands out as the leading mode and has dramatic impacts on global





and regional climates (Ropelewski and Halpert, 1986, 1987, 1989, 1996; Kiladis and Diaz, 1989; Harrison and Larkin, 1998; Dai and Wigley, 2000; McCabe and Dettinger, 1999; Wang et al., 2000). In recent years, discussion about non-canonical ENSO (e.g., Central-Pacific El Niño or El Niño Modoki, Ashok et al., 2007) and associated impacts has stimulated much more ENSO-related research. Since ENSO has been studied for decades, for quantitative and monitoring purposes, researchers have

defined several ENSO indices, including NINO 1+2, NINO 3, NINO 4, and NINO 3.4 (e.g., Trenberth, 1997; Trenberth and Stepaniak, 2001).

In addition to ENSO, plentiful teleconnection patterns over fields of atmospheric variables (e.g., sea level pressure and 500-mb geopotential height) have been recognized by scientists using various statistical techniques, such as correlation analysis by Wallace and Gutzler (1981) and rotated principal component analysis (RPCA) by Barnston and Livezey (1987). Prominent

teleconnection patterns include the North Atlantic Oscillation (NAO, Hurrell, 1995), Pacific North American (PNA, Wallace and Gutzler, 1981; Barnston and Livezey, 1987), Indian Ocean Dipole (IOD, Saji et al., 1999), West Pacific (WP, Barnston and Livezey, 1987), East Pacific-North Pacific (EP-NP, Barnston and Livezey, 1987), Pacific-Japan (PJ, Nita, 1987; Kosaka and Nakamura, 2006), Artic Oscillation (AO, Thompson and Wallace, 1998), and Antarctic Oscillation (AAO, Gong and Wang, 1999). If a high-frequency filter is applied to atmospheric/oceanic variables of interest (i.e., to reserve the low-frequency

component) prior to pattern recognition, other patterns referred to as "interdecadal modes" can be observed. For example, the Pacific Decadal Oscillation (PDO, Mantua et al., 1997) and Atlantic Multidecadal Oscillation (AMO, Schlesinger and Ramankutty, 1994; Enfield et al., 2001) both shift phases with a period longer than several decades. All the above interannual and interdecadal oscillations have shown widespread impacts on the regional and global climate system.

East Asia is a noticeable geographical region modulated by a variety of weather systems in different seasons and has been

greatly affected by teleconnection patterns, as witnessed by many studies (e.g., Wang et al., 2000; Yang et al., 2002; Wang and Fan, 2005; Choi et al., 2012). Taiwan features most weather systems found in East Asia, including spring rains, Mei-Yu, and East Asian monsoon from spring to summer, typhoons from summer to autumn, and the Mongolian high pressure system and associated northeast monsoon in winter. Therefore, seeking the relationship between Taiwan's climate and large-scale circulations can provide some clue to dissect the mechanisms of East Asian climate. This prompts our investigation of

incorporating teleconnection patterns into a predictive framework for hydro-climatic variables in Taiwan. If such framework can be shown effective, it can then open the door for applications in other East Asian regions.

Using teleconnection patterns to predict regional precipitation is straightforward as precipitation is such a hydro-climatic variable signifying the regional sink of large-scale moisture transport. However, precipitation forecasting could be largely hampered by its spatio-temporal heterogeneity, mainly attributed to the influence of rugged landform (as the characteristic

topography in Taiwan, Lee et al., 2015). Because of the interaction between topography and distinct synoptic weather systems among seasons, the challenge of precipitation forecasting in Taiwan can easily escalate. On the other hand, streamflow data measured at a watershed outlet represent an *integrated* response to spatial and temporal precipitation distribution within the watershed, largely attenuating the influence of precipitation heterogeneity. In addition, given the assumption that evapotranspiration could be neglected during the wet season (e.g., summer in Taiwan), the streamflow variable can be an ideal

surrogate for interpreting long-term climate variability. The explicit linkage between streamflow and water resources and



hydro-meteorological hazards also indicates that effective streamflow prediction can produce immediate utility for a variety of users.

In fact, the usage of teleconnection patterns for streamflow forecasting is enlightened by a number of previous endeavors. Earlier work can be exemplified by Kahya and Dracup (1993) and Hamlet and Lettenmaier (1999); whereas the former examined the relationship between ENSO and unimpaired streamflow over the contiguous United States, the latter devised an empirical model for the forecasting of Columbia River streamflow using ENSO and PDO. Another earlier work by Chiew et al. (1998) was to link ENSO to Australian streamflow, and then Chiew and McMahon (2002) later extended their discussion to global ENSO-streamflow teleconnection. More recently, Moradkhani and Meier (2010) adopted various climate indices along with several climate variables and then employed a principal component regression model for streamflow forecasting in two Pacific Northwest basins; Robertson and Wang (2012) applied a Bayesian approach to select predictors from a pool of 13 climate indices for seasonal streamflow forecasting in Australia; Hidalgo-Muñoz et al. (2015) used multiple linear regression with 20 teleconnection indices as potential predictors to forecast seasonal streamflow over the Iberian Peninsula. From these studies, it is noted that while the usage of a comprehensive list of climate indices appears to be the trend, most studies focused on coping with modern statistical techniques and pursuing optimal skill rather than diagnosing underlying mechanisms of predictability and pointing caveats on intrinsic covariability between regional streamflow and large-scale circulations. Furthermore, to the best of our knowledge, similar work applied to Taiwan or most East Asian regions has not been conducted yet. The above rationale therefore serves as the backbone of our research opportunity.

One of the caveats on using teleconnection patterns for hydro-climatic forecasting that should be addressed is the existence of climate regime shifts (CRS). The climate system, as demonstrated by the phase changes in the PDO and AMO, can undergo a reconfiguration shifting its state from one to another. A steady climate regime can last for decades, but a CRS usually takes place momentarily (e.g., in a particular year or so). The occurrence of a CRS indicates not only a new climate state but also a deterioration or even a possible break-off of the relationship between regional hydro-climates and certain circulation patterns. Many researchers have identified some notable CRS in the Pacific basin (e.g., the late 1970s by Miller et al., 1994; Hare and Mantua, 2000, and the late 1990s by McPhaden et al., 2011; Hong et al., 2014a). However, neither the shift in the correlation of Taiwan's streamflow with teleconnection patterns nor the impact of the CRS on streamflow forecasting has been thoroughly discussed.

The above introduction explains the main motivation for our work, which comprises of two primary objectives:

1. To investigate the relationships between Taiwan's streamflow and teleconnection patterns through conducting correlation analysis between seasonal streamflow data and major climate indices; and,

2. To examine and verify the existence of any CRS signals in the correlation and to discuss how associated changes in large-scale circulation patterns may influence the potential application for seasonal forecasting.

The rest of this paper is organized as follows. Section 2 describes data and analysis procedures used in this study. To align with the study objectives, Sections 3 and 4 carry out the results of correlation analysis and the discussion of CRS and implication for seasonal forecasting, respectively. Lastly, Section 5 provides a summary of our findings and concluding remarks.





## 2 Data and Analysis Procedures

This study focuses on discussing the relationships between Taiwan's streamflow and large-scale circulations. Taiwan is an island country located at East Asia around 23.5° N and 121° E. The major island of Taiwan (∼99% of its territory) has an area about 36,000 km$^2$. As mentioned earlier, Taiwan's climates are modulated by various weather systems in four seasons.
Because of the Central Mountain Range (topographic variations) and gradually varied climate zones (latitudinal differences), the influence of those weather systems on precipitation can show great east-west and north-south contrasts. As a result, while the wet season generally spans from summer to autumn based on the long-term average, Taiwan's streamflow in the wet season exhibits great spatial distributions of prominent intra-seasonal and inter-annual variations. Figure 1 illustrates the location of Taiwan and highlights those major watersheds and sub-catchments analyzed in this study. The streamflow data in the watersheds
will be correlated with selected climate indices. In the subsections below, the specifications and sources of the streamflow, climate index, and other auxiliary data will be amply described, followed by the analysis procedures used in this study.

### 2.1 Streamflow data, climate indices, and other data sets

Streamflow data used in this study are obtained from the Water Resources Agency in Taiwan, the primary authority in charge of installing and monitoring most of the river gauges over the country, and from the Taiwan Power Company, measuring
streamflow for the sake of hydroelectricity. Out of many gauges possessed by these agencies, a total of 28 upstream and 13 downstream gauges of satisfactory quality and extended record is selected. The collective contributing area associated with those downstream gauges located at the outlets of 13 major watersheds is about 16,731 km$^2$, ∼46% of Taiwan's territory. Since streamflow data observed at the downstream gauges are subject to human intervention (e.g., water regulation and withdrawal for various consumption), a distinction has been made to separate those upstream gauges with pristine flows from the downstream
gauges, as shown in Figure 1. From here onward, two batches of the same (or similar) analysis will be performed for the upstream and downstream data. However, to minimize the effect of human intervention on streamflow data (and to still include those downstream data in our analysis), the scope of this study is designed to emphasize data during the high-flow season (as frequent regulations usually take place during low-flow seasons). Without loss of generality, July to September (JAS) are identified as the target high-flow season, from which seasonal streamflow data are aggregated for our follow-up analysis. JAS
is also known as the major typhoon season in Taiwan, so our analysis will present implication in reflecting typhoon activity to some extent. Here the simplest strategy is adopted for avoiding human intervention, but some alternatives to reconstructing natural flows (e.g., Wen, 2009) do exist and could facilitate our ongoing analysis during other seasons.

JAS streamflow data thus far prepared for the target upstream and downstream gauges represent potential predictands for seasonal forecasting, and this study plans to analyze how these predictands correlate with the major teleconnection patterns.
This correlation analysis is purposeful since the major teleconnection patterns can emulate a miniature of the full-scale climate system for a great deal of climate variability explained, thereby providing some clues about the causations of the interannual variations of Taiwan's streamflow. A list of the major teleconnection indices has been compiled for this purpose. The desired list should cover as many teleconnection patterns as possible, but those selected should show certain signs of connections to



East Asian climate based on previous literature. As the well-known impact of ENSO on East Asian climate, the list begins with three ENSO indices, namely NINO 1+2, NINO 3.4, and NINO 4, which represent the sources of influence from East, East-Central, and Central Tropical Pacific, respectively. In the immediate vicinity of the Pacific, the Indian Ocean has the IOD as the leading mode with evidence of steering East Asian Monsoon and other weather systems (Guan and Yamagata,

2003), so the IOD index is selected. Over the farther side and higher latitude of the Pacific, the EP-NP and PNA patterns, found to be associated with the intensity and location of the Pacific jet stream (e.g., Yang et al., 2002), are included in the list. Although sprung from the Polar Regions, the AO and AAO were indicated by more and more studies that they can affect climate variability in remote, subtropical regions (e.g., Wang and Fan, 2005; Choi et al., 2012); it would be reasonable to have these two indices in the list. To account for possible transoceanic interactions recently addressed by some studies (Hong et al.,

2014b), the NAO index is also included in the list. Furthermore, as the predictand of interest is highly related to summertime tropical cyclone (TC) activity, the list contains the QBO (Quasi-Biennial Oscillation, Baldwin et al., 2001), WP, and PJ indices (Choi et al., 2010; Kosaka et al., 2013). Beyond the above climate indices, the PDO and AMO indices, as the representative of the interdecadal oscillations, are also included in the list for examining any low-frequency connections. Table 1 displays the list of the climate indices and depicts their sources as well as some key references.

To reveal large-scale patterns pertaining to certain climate indices (or even unprecedented signals) that potentially dominate Taiwan's streamflow variability, this study uses some other data sets including the Extended Reconstructed Sea Surface Temperature (ERSST Version 4, Huang et al., 2015), NCEP/NCAR Reanalysis Sea Level Pressure (SLP, Kalnay et al., 1996), and the Global Precipitation Climatology Project (GPCP Version 2, Adler et al., 2003).

## 2.2 Lagged correlation analysis

The design of the correlation analysis is explained as follows. Firstly, correlation coefficients (Pearson's $r$ is used) between the JAS flow data at one of the gauges and different climate indices in the same season (e.g., JAS ENSO index) are calculated. The calculation of the "concurrent" correlations is then repeated until all the gauges are covered. Secondly, since the motivation of this work is to explore any forecasting possibilities, lagged correlations are computed as well. Lagged correlations are calculated between the JAS flow data and the climate indices averaged over the preceding three-, six-, and nine-month seasons,

namely, AMJ, JFMAMJ, and ONDJFMAMJ, respectively. This approach is commonly adopted by plentiful forecasting studies (e.g., Rajeevan et al., 2007) as the extension of averaged periods can eliminate some high-frequency or artificial data disturbance. While averaged over as many as nine preceding months, the lag correlations provide prediction skills only at a lead time of one month. Prediction skills at longer lead times could be obtained from calculating lagged correlations between the JAS flow data and the climate indices averaged over other preceding seasons (e.g., JFM, OND, and even previous JAS). This part

of effort will be continued in another working paper. The above correlation analysis is applied to all the climate indices with the exception of the PJ index. Since the PJ index supplied by the source (Kubota et al., 2016) is derived from the JJA data, only the semi-concurrent correlations (i.e., JAS flow data vs. JJA PJ index) can be calculated for this index.





### 2.3 Climate regime shift test

If Taiwan's streamflow exhibits significant correlations with certain climate indices, our ensuing analysis is to examine whether such correlations show any sign of temporal disruption, indicating a possible climate regime shift. The temporal disruption or abrupt changes in a univariate time series can be commonly identified by using classic, nonparametric techniques, such as

the Mann-Whitney-Pettitt (MWP, Pettitt, 1979) and Kruskal-Wallis (KW, Kruskal and Wallis, 1952) tests. However, these techniques may not be directly applicable to quantities like temporal correlations derived from a bivariate time series. A simple resolution of the change-point identification under this circumstance would be the usage of moving-window approach to obtain a "correlation time series" to which either the MWP or KW test can be applied. Another statistically sound approach to this problem, proposed by Rodionov (2015), is used in this study. Rodionov's method in detecting abrupt changes in the correlation

coefficient is based on the fundamental property of variance. If there are two variables of interest, $x$ and $y$ (e.g., streamflow and climate index), the variance of the sum of $x$ and $y$ can be written as:

$$S^2_{x+y} = S^2_x + S^2_y + 2rS_xSy, \tag{1}$$

where $S^2$, $S$, and $r$ denote the sample variance, standard deviation, and correlation coefficient, respectively. Further, if the two variables are normalized (i.e., $x = y = 0$ and $S^2_x = S^2_y = 1$), the above equation can be reduced to:

$$S^2_{x+y} = 2(1+r). \tag{2}$$

Note that since the sample correlation coefficient ranges from -1 to 1, the above variance is bounded between 0 and 4. Equation 2 also indicates that the identification of shifts in $r$ is equivalent to that in $S^2_{x+y}$. In his previous work, Rodionov (2005) has introduced a method for detecting the abrupt shifts in the variance (of a single variable) based on a "sequential $F$-test." Therefore, the same method for the variance of $x$ or $y$ can be simply applied to the variance of $x + y$, thereby achieving the

identification of shifts in the correlation coefficient. In essence, the above method can also be applied to the variance of $x - y$,

$$S^2_{x-y} = 2(1+r), \tag{3}$$

which should theoretically yield very similar change-points if the $p$-value (computed from the Fisher's $r$-to-$z$-transformation, Fisher, 1921) is less than 0.05 (i.e., a high significance level). As per Rodionov's suggestion, the test is performed for both the sum and difference series to ensure the minimum $p$-value is attained.

If the variables have not been normalized, Rodionov (2015) stated that some pre-processing work on the time series is required: using the shift detection in the mean (based on a "sequential $t$-test," Rodionov, 2004) and variance to obtain the stepwise means/trends and variances, respectively. The variables can then be normalized for the shift detection in the correlation coefficient. In addition to the pre-processing work and the desired significance level, a cut-off length $l$ should be determined for the method to detect change-points and to calculate associated statistics. For more information about the suite of the CRS

detection methods, please refer to Rodionov's previous studies.



## 3 Results

### 3.1 Correlations between Taiwan's runoff and climate indices

In line with the aforementioned instruction, the correlation analysis is conducted for all the target gauges in Taiwan. Since the total number of combinations of the different gauges (upstream and downstream), climate indices, and lagged periods is in the thousands, the resulting correlation values are merely too many to be fully listed here. Therefore, the results are presented in a selected and illustrative fashion. In Figures 2 and 3, concurrent and lagged correlations between the JAS runoff at the upstream gauges and selected climate indices are color coded over the maps of the catchments. The criterion for selecting climate indices is that any of them exhibit correlation values at the 95% confidence level with at least one of the catchments under either the concurrent or lagged scenario. In other words, those climate indices excluded from the plots do not show significant concurrent or lagged correlations with none of the upstream catchments. In addition, please note that the lagged correlations shown in Figure 3 are based on the average period over ONDJFMAMJ only since lagged correlations based on other average periods (see Section 2.2) show quite similar patterns but less significant results. In the two figures, the highest absolute correlation value among all the upstream catchments for a specific climate index is also denoted in each plot.

Several key messages can be deciphered from Figures 2 and 3. Under the concurrent scenario: 1) Many upstream catchments show significant positive correlations with the climate indices; among these indices, the WP and PJ indices stand out to have the strongest, universally in-phase relationship with the JAS runoff, indicating the direct influence of typhoon activity during the same season. 2) The only climate index showing all negative correlations with the upstream catchments is the PDO; this out-of-phase relationship is supported by some existing findings (e.g., Li et al., 2010) regarding the PDO's influence on East Asian climates. 3) Surprisingly, the NAO, as the farthest teleconnection mode, shows quite strong correlations (∼99% confidence level) with some upstream catchments. On the other hand, under the lagged scenario: 1) In comparison with the concurrent correlation patterns, some of them experience a clear phase reversal over certain regions of Taiwan (e.g., North-Northeast for IOD and WP and Central-Southwest for PNA). 2) The most pronounced out-of-phase relationship is observed from the QBO index, and this relationship is even stronger than any of the concurrent relationships.

To further interpret the range of correlation values, Figure 4 encapsulates the correlations of the JAS runoff at every gauge against every climate index using box-and-whisker plots. Each box plot constituted by 28 (13) values represents the range of correlations observed from the upstream (downstream) gauges. Results for upstream and downstream gauges were separately illustrated at the left and right panels, respectively. Concurrent and lagged correlations with JAS (JJA for PJ) and ONDJFMAMJ climate indices are both shown at the top and bottom of Figure 4, respectively. Besides, the abscissa of Figures 4a and 4b is ranked by the mean correlation in each box plot for a clearer illustration of the performance between climate indices. To observe whether a phase transition takes place, the same ranking is then inherited by Figures 4c and 4b. In addition, the corresponding correlation values for all the downstream gauges are also enumerated in Table 2 (values for the upstream gauges are available upon request).

Compare Figure 4a with 4b, the ranking of climate index is nearly the same (except PNA), and the interquartile range (IQR) of each box plot is also similar, implying the general scale (of catchments) consistency in response to large-scale circulations



and the little influence of human intervention on the JAS runoff for the downstream. However, the total ranges of correlation values are wider for the upstream, which possibly reflects the higher randomness of catchments at a smaller scale. Under the concurrent scenario, out of the 14 climate indices, ten and nine tend to positively correlate with the JAS runoff for the upstream and downstream, respectively. While under the lagged scenario (Figures 4c and 4d), only four climate indices are positively

correlated with the JAS runoff for both the upstream and downstream gauges (i.e., the phase reversal mentioned earlier). In fact, the PNA, IOD, NINO4, NINO3.4, and WP are the climate indices showing the phase reversal for both the upstream and downstream JAS runoff.

## 3.2  Regime shifts in the correlation

Although some of the correlation results for certain catchments against climate indices disclosed above are at a moderate

significance level, it is found that these correlations are subject to peculiar interannual variations. For instance, 20-year moving-window correlations with four climate indices, namely the PDO, EPNP, AO, and PJ, are plotted in Figure 5. In the figure, each box plot presents all correlation values pertaining to 28 upstream catchments between the 20-year JAS runoff and the specific climate index. The year label indicates the end of a moving window (e.g., 1989 indicates the moving window is taken from 1970 to 1989). Blue and magenta time series in each plot represent two catchments resulting in the highest and lowest moving-

window correlations over the temporal horizon, respectively. These two time series can go outside the box plot range (i.e., deemed an outlier at a specific time point) since the IQR with a factor of 1.5 is used to determine a rather diminished range. From these plots, several types of interannual variations as potential signs of CRS can be identified: 1) divergence of the total ranges along with time (e.g., Plots a, b, and c), 2) a gradual decreasing trend (e.g., Plot b), 3) an abrupt increase or decrease (e.g., Plots c and d), and 4) a change of sign in the mean correlation near the suspicious CRS year (e.g., Plots b and c). Other

moving-window sizes (e.g., 21 years as adopted by Hung et al., 2004) have been tested as well, and the results are very similar to (but less pronounced than) those presented herein.

As stated in the moving-window correlations, we hypothesized that the relationship between Taiwan's streamflow and large-scale circulations may have undergone several climate regime shifts. To validate this hypothesis and to identify possible years of CRS, the CRS test is applied to the streamflow data and climate indices. Based on the spatial significance of the results of

correlation analysis, the aggregated JAS runoff for East (West) Taiwan is computed from the average of HLI and SGL (WU, JS, and BG) data for the CRS test. Figure 6 presents the result of the CRS test on identifying any shifts in the correlation between the East and West Taiwan runoff versus the PJ index. The result indicates two highly significant change points in 1979 and 1999 (significant at $p = 0.0001$ and $0.0003$, respectively) for the East Taiwan runoff. Between 1979 and 1999, the correlation value is almost 0.9, suggesting the dominant effect of the PJ pattern on the moisture transport to Taiwan in summer. Such strong

correlation is hardly seen in the field of climate sciences; however, the correlation deteriorates drastically after 1999, becoming slightly negative and close to zero. Likewise, two very significant change points in 1988 and 2000 (significant at $p = 0.006$ and $0.0003$, respectively) can be identified for the West Taiwan runoff. While the first change point is identified in the late 1980s, the marked in-phase relationship between 1979 and 1999 is still quite notable. The same CRS test has been applied to Taiwan's runoff versus all the other climate indices, but only less significant results can be found.



# 4 Discussion

In this section, further discussion is made to address the issue of CRS and how it can impact the convention of seasonal forecasting evidenced by some large-scale patterns. To start with the discussion, we would like to argue that the two change points found for the response of Taiwan's streamflow to large-scale circulations are not a fluke. Whereas the first change point in the late 1970s (for the East Taiwan runoff in particular) is clearly in relation to the widely known CRS identified over the entire Pacific basin SST (Miller et al., 1994; Hare and Mantua, 2000), the second change point in the late 1990s coincides with the CRS induced by a warming over the equatorial western Pacific (McPhaden et al., 2011; Hong et al., 2014a) and/or more frequent occurrences of the central Pacific El Niño (Xiang et al., 2013). To supplement our explanation here, the CRS analysis of shifts in the mean is also applied to all the climate indices examined in this study, and the change points corresponding to each climate index are listed in Table 3. From the table, the shifts in the mean identified for the PDO are very much consistent with previous studies, and only such shifts (rather than the PJ) are in perfect agreement with the identified shifts in the correlation for Taiwan's streamflow. Therefore, our hypothesis of the identified CRS can be drawn as follows: The CRS firstly emanates from the change in the basin-scale climatology over the Pacific (e.g., shift in the PDO), and then the reorganized large-scale patterns can reset the relationship between the island-scale streamflow with established regional circulations (e.g., the PJ pattern).

The observation of the CRS can profoundly influence predictor screening for empirical forecasting methods. Conventional predictor screening usually relies on pattern correlation (Chen and Georgakakos, 2014); that is, identifying specific areas over a predictor field (usually SST for its considerable energy absorption and slowly varying property) showing significant correlation with the predictand of interest (e.g., local precipitation or streamflow) over a sufficiently long period of time. This concept holds true if the predictor-predictand relationship remains quasi-stationary. On the contrary, if the predictor-predictand relationship is no longer stationary (e.g., identification of prominent CRS in the correlation), this concept of predictor screening would become rather questionable. To illustrate the impact of the CRS on seasonal forecasting, the PJ pattern, which shows overall the highest concurrent correlation with the JAS runoff in Taiwan over a specific time window, is used to generate some large-scale patterns next.

Firstly, two sets of pattern correlation of the PJ index with the SLP, SST, or GPCP data are generated using 1999 as a demarcation of the time window (Figure 7). Before 1999, the PJ-SLP correlation indicates a marked dipole pattern with its two poles centered at southern Japan and near Taiwan. Along with the significant negative correlation (at $p = 0.05$) extended to the Indian Ocean (regarding SST) and the positive correlation over Taiwan (regarding GPCP), all these patterns reasonably reproduce the canonical PJ pattern defined by previous studies (e.g., Kubota et al., 2016). In contrast, from 1999 onward, the entire set of pattern correlation alters dramatically: The dipole pattern for the PJ-SLP correlation becomes less significant, and the extensive negative correlation over the Indian Ocean for the PJ-SST correlation reverses sign. In addition, the positive correlation originally occupying the Taiwan area migrates northeast to East China Sea. The above analysis implies that owing to a possible CRS, a specific climate index (e.g., PJ) can respond quite differently to large-scale circulations from one time window to another. More importantly, if that climate index is adopted as a predictor (often seen in numerous long-lead forecasting





applications, e.g., Moradkhani and Meier, 2010; Robertson and Wang, 2012; Hidalgo-Muñoz et al., 2015), its relationship with the predictand (e.g., precipitation over Taiwan) could largely weaken or even terminate given the existence of a CRS.

If pattern correlation is calculated directly between the East Taiwan runoff (as the predictand) and the SLP, SST, or GPCP data (as the predictor fields), the bulk of significant areas in the predictor fields usually specifies the spatial extent of potential

predictors, in accordance with the common procedure of predictor screening appearing in many articles (DelSole and Shukla, 2009). Figure 8 demonstrates this procedure twice, one before and the other after the identified CRS. Obviously, the left column of Figure 8 resembles that of Figure 7 to a very high degree, indicating the PJ pattern is indeed an ideal predictor candidate before 1999. Nevertheless, from 1999 onward, the right column of Figure 8 resembles neither the left column of the same figure nor the right column of Figure 7, implying that the PJ predictor should be replaced by some other predictor candidates

(e.g., SST over the subtropical central Pacific).

Lastly, to reach some forecasting utility, lead time information can be incorporated into the procedure of predictor screening. For instance, Figure 9 presents the evolution of the SST anomaly (SSTA) composites based on wet-minus-dry years of the JAS runoff for East Taiwan, which can also reveal significant predictor areas at varied lead times (Chen and Georgakakos, 2015). The identified CRS is used to divide the generation of the SSTA composites. Before the CRS, (1982, 1984, 1985, and 1994) and

(1980, 1983, 1993, and 1998) are identified as the wet and dry years, respectively. After the CRS, (2001, 2005, and 2007) and (1999, 2002, and 2010) are identified as the wet and dry years, respectively. Wet and dry years are identified as those years for which the JAS runoff is higher and lower than the $67^{th}$ and $33^{rd}$ percentiles of the entire data series, respectively. During the concurrent season, notwithstanding that Figures 9(a) and (b) show the SSTA patterns as the extension of the patterns observed in Figures 8(c) and (d), it can be found that the change in the patterns before and after the CRS is significant up to the global

scale. With increases in lead times, the two sets of SSTA composites present distinct evolutionary pathways. Eventually, at the 9-month lead time, the SSTA composite expresses a La Niña pattern for the before-CRS scenario, completely opposite to an El Niño pattern for the after-CRS scenario. Overall, the SSTA composites imply that a predictor screening scheme will be effective only if it can anticipate a forthcoming CRS; otherwise, it would be no surprise that some busted forecasts could be made periodically.

## 5 Concluding Remarks

A set of teleconnection patterns is a proxy for the complex climate system, and these patterns have shown dominant effects on modulating regional hydro-climates in different seasons. A response of a region's hydro-climates to the global climate system can be characterized by the regional sink of moisture transported via various circulations and mechanisms. Along this train of thought, this study attempted to shed light on the relationship between teleconnection patterns and Taiwan's streamflow. It

is believe that streamflow data could be a more descriptive and preferable metric than precipitation relating to teleconnection patterns for its sustained (i.e., less transient), integrated water response to the climate system and direct association with water resources and hydro-meteorological hazards. Long-term JAS streamflow data derived at 28 upstream and 13 downstream gauges in Taiwan were used to correlate with 14 teleconnection indices showing signs of linkage to East Asian climate. To





scrutinize potential forecasting utility, the climate indices were averaged over not only JAS but also the preceding seasons for the calculation of concurrent and lagged correlations. In the course of our correlation analysis, it was noted that some significant correlation results based on the entire period of record were actually induced by an even more significant in-phase (or out-of-phase) relationship during a truncated time frame, indicating inherent climate regime shifts. Therefore, CRS analysis was performed to identify any significant shifts in the correlation as well as the mean of the climate indices themselves. Discussion of how the found regime shifts impact empirical prediction was then carried out. Our key findings are summarized below.

1. Among those many teleconnection patterns, we found that the WP and PJ patterns exhibit the highest concurrent correlations with the JAS flows at both the upstream and downstream catchments in Taiwan. The determinant of such correlation performance should be the association of these patterns with cyclonic activity in the western North Pacific basin.

2. On the other hand, the lagged correlation analysis indicated that the QBO index at a one-month lead time significantly correlate with the JAS flows in Taiwan, promoting its forecasting utility. In addition, some of the climate indices change their relationships with the JAS flow from mostly positive concurrent correlations to negative lagged correlations.

3. We identified two shifts in the correlation between the PJ index and Taiwan's streamflow in the late 1970s and 1990s, and then verified that these shifts should be bonded to some existing findings of the alteration of large-scale circulations in the Pacific around the same time.

4. Pattern correlation and composite analysis further illustrated drastic changes of large-scale patterns before and after the change point in the late 1990s. Our results overturn the convention of predictor screening widely used for empirical forecasting as the alignment of predictors can vary in consonance with certain interannual and interdecadal oscillations.

Our current endeavor includes applying the similar analysis framework to Taiwan's streamflow in other seasons. Once clear evidence of climate regime shifts in the succeeding work can be gathered, a new predictor screening algorithm capable of accounting for CRS will be developed. This algorithm will be incorporated into a seasonal streamflow forecasting model for improving water resources planning and management in East Asia.

*Author contributions.* C-J. and T-Y. designed and conducted the analysis and co-wrote the manuscript.

*Acknowledgements.* Work by C-J. Chen was supported by Taiwan's Ministry of Science and Technology under grant: MOST 104-2625-M-005-007-MY2. Work by T-Y. Lee was also supported by Taiwan's Ministry of Science and Technology under grant: MOST 104-2116-M-003-005. The authors express the upmost gratitude to Dr. Hisayuki Kubota for providing the PJ index and the Water Resources Agency and Taiwan Power Company for providing the streamflow data.





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



**Table 1.** List of 14 climate indices used in this study.

| Climate Index | Full Form (or Definition) | Source | Sample Reference |
|---|---|---|---|
| AMO | Atlantic Multidecadal Oscillation | NOAA/ESRL/PSD | Enfield et al. (2001) |
| PDO | Pacific Decadal Oscillation | jisao.washington.edu | Mantua et al. (1997) |
| NINO1+2 | ENSO index at East Equatorial Pacific | NOAA/NCEP/CPC | Trenberth (1997) |
| NINO3.4 | ENSO index at East-Central Pacific | NOAA/NCEP/CPC | Trenberth (1997) |
| NINO4 | ENSO index at Central Pacific | NOAA/NCEP/CPC | Trenberth and Stepaniak (2001) |
| IOD | Indian Ocean Dipole | jamstec.go.jp | Saji et al. (1999) |
| EPNP | East Pacific-North Pacific | NOAA/ESRL/PSD | Barnston and Livezey (1987) |
| PNA | Pacific North America | NOAA/ESRL/PSD | Wallace and Gutzler (1981) |
| AO | Artic Oscillation | NOAA/NCEP/CPC | Thompson and Wallace (1998) |
| AAO | Antarctic Oscillation | NOAA/ESRL/PSD | Gong and Wang (1999) |
| NAO | North Atlantic Oscillation | NOAA/ESRL/PSD | Hurrell (1995) |
| QBO | Quasi Biennial Oscillation | NOAA/ESRL/PSD | Naujokat (1986) |
| WP | West Pacific | NOAA/ESRL/PSD | Barnston and Livezey (1987) |
| PJ | Pacific Japan | From Dr. Hisayuki Kubota | Kubota et al. (2016) |



**Table 2.** Results of correlation analysis for the major watersheds in Taiwan; values before (after) the slash are concurrent (lagged) correlation coefficients ($\times 10^{-2}$; significant at $p = 0.05$ are bold and italic).

| Watershed† | AMO | PDO | NINO1+2 | NINO3.4 | NINO4 | IOD | EPNP | PNA | AO | AAO | NAO | QBO | WP | PJ‡ |
|---|---|---|---|---|---|---|---|---|---|---|---|---|---|---|
| TC (NW) | 1/0 | -10/1 | -11/-15 | 21/-9 | *25/1* | 2/-9 | -1/7 | -6/-12 | 4/0 | 3/-4 | *28/8* | 0/-20 | 10/-9 | *25/** |
| HLO (NW) | 29/24 | -28/-17 | -23/-32 | 13/-23 | 16/-11 | -7/-9 | -23/5 | 9/-20 | 5/-7 | 14/7 | 16/-5 | -2/*-34* | 26/-5 | 32/* |
| WU (W) | -14/-24 | 4/0 | -5/-16 | 11/-11 | 6/-9 | 11/-10 | 15/2 | -12/-13 | 3/11 | 0/2 | 22/26 | -2/-5 | *37/0* | *45/** |
| JS (W) | 17/8 | -22/-17 | -5/-27 | 8/-30 | 4/-25 | 11/-4 | -9/-19 | 24/-18 | -3/-10 | 8/24 | 12/3 | -13/-28 | *36/-1* | *33/** |
| BG (W) | -3/-7 | *-27/-6* | -12/-14 | 0/-13 | -6/-19 | 6/-8 | 11/-8 | -7/-4 | -7/-22 | 8/-3 | *25/-14* | 3/-21 | *31/13* | 26/* |
| ZW (SW) | 8/3 | *-27/-28* | 4/-19 | 11/-19 | 5/-17 | 11/-6 | -5/-16 | 15/-16 | -3/7 | -3/-1 | 13/-7 | 1/*-27* | 19/-9 | 16/* |
| ER (SW) | 9/15 | -22/-9 | -6/-15 | -6/-15 | -5/-12 | 18/3 | -24/-10 | 10/-4 | 7/5 | -15/10 | 12/6 | -19/-6 | 22/-9 | 5/* |
| GP (SW) | -1/-2 | *-26/-14* | -9/-21 | 7/-20 | 9/-17 | 16/-5 | -2/-4 | 19/-15 | -4/-10 | -8/15 | *26/8* | -3/-22 | *25/8* | *30/** |
| BN (SE) | 18/16 | -10/-21 | -6/-24 | 15/-11 | 12/-7 | -1/7 | 5/-5 | 6/-11 | -7/-12 | -16/18 | -4/-15 | -5/-10 | 5/-15 | 9/* |
| SGL (E) | -5/-6 | -13/0 | -19/-25 | 10/-9 | 16/-3 | 3/-6 | 8/-7 | -8/-10 | 6/4 | 2/8 | 15/8 | -16/*-35* | 20/-17 | *41/** |
| HLI (E) | 26/21 | *-36/-21* | -19/-27 | 7/-9 | 14/1 | 12/5 | -25/-14 | 17/-15 | 9/-8 | -2/11 | 7/-12 | -12/*-38* | 5/0 | 20/* |
| HP (NE) | -9/7 | -7/-3 | -23/-17 | -3/2 | 5/10 | 0/2 | 0/-2 | 15/-18 | 8/11 | -26/-25 | 20/4 | -7/-30 | 20/-26 | 7/* |
| LY (NE) | 13/19 | -19/-6 | 2/-8 | 21/-5 | *26/-1* | 7/11 | -12/1 | 9/-1 | -5/-13 | -7/-11 | -10/-4 | -18/*-44* | -2/-4 | 9/* |

†: Inside the parenthesis is the relative location of each watershed in Taiwan (NW: NorthWest; W: West; SW: SouthWest; SE: SouthEast; E: East; NE: NorthEast).

‡: Lagged correlation is not available for the PJ index.





**Table 3.** Change points (shifts in the mean) identified for all the JAS climate indices examined in this study; significance level is reported by the number of asterisks (*: $p \leq 0.05$; **: $p \leq 0.01$; ***: $p \leq 0.001$; ****: $p \leq 0.0001$).

| Climate Index | Change Points |
|:---:|:---:|
| AMO | 1962 (****), 1987 (*), 1998 (***) |
| PDO | 1976 (****), 1998 (**), 2010 (**) |
| NINO1+2 | None |
| NINO3.4 | None |
| NINO4 | 1990 (****) |
| IOD | 2006 (**) |
| EPNP | 1997 (****) |
| PNA | 2002 (**) |
| AO | 1982 (*) |
| AAO | None |
| NAO | 1967 (*), 2006 (*) |
| QBO | None |
| WP | 2002 (*) |
| PJ | 1972 (*), 1991 (*) |





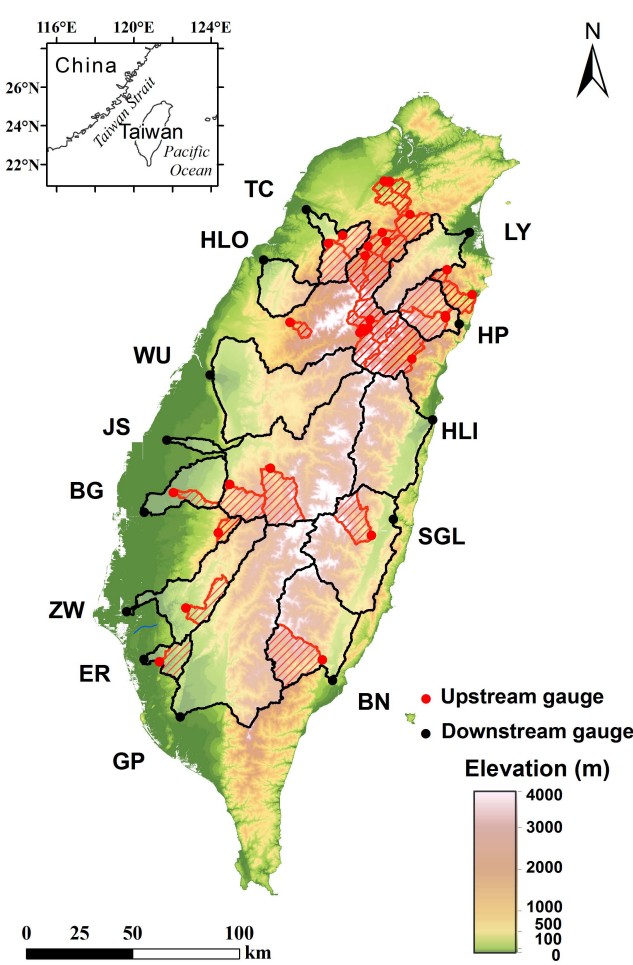

**Figure 1.** 13 major watersheds (black boundaries with abbreviations near the outlets) of Taiwan and 28 upstream catchments (red shadings) analyzed in this study.





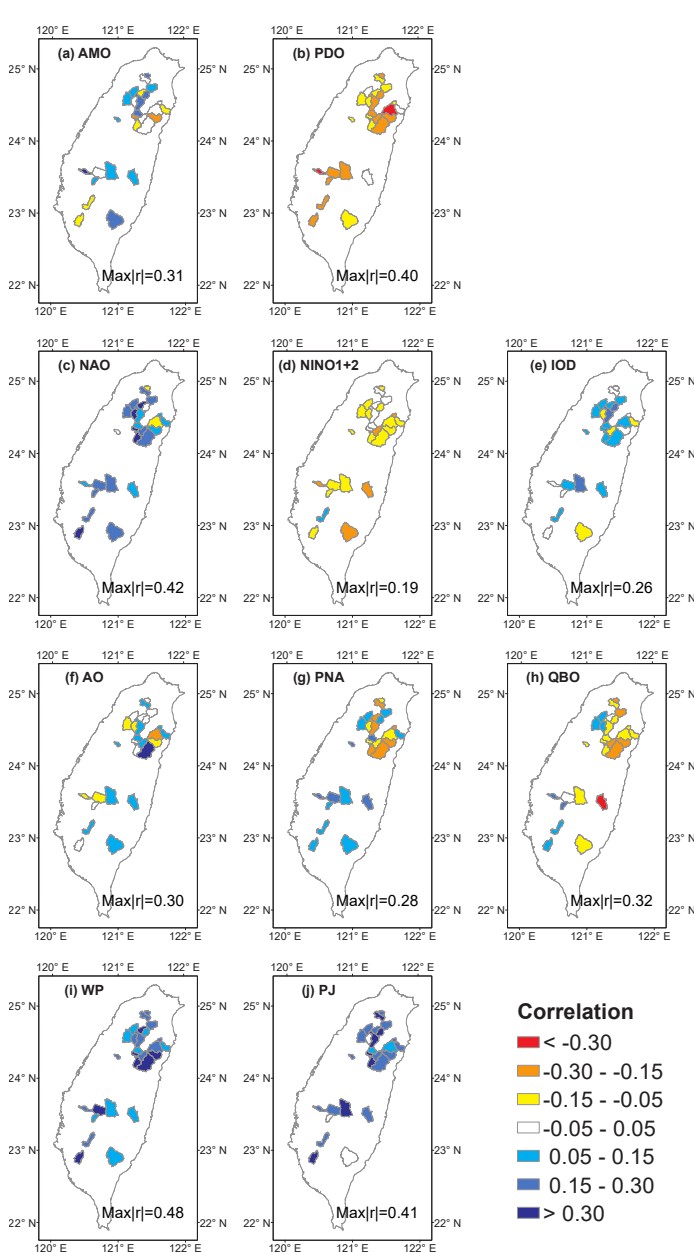

**Figure 2.** Maps of concurrent correlations showing at each upstream catchment.





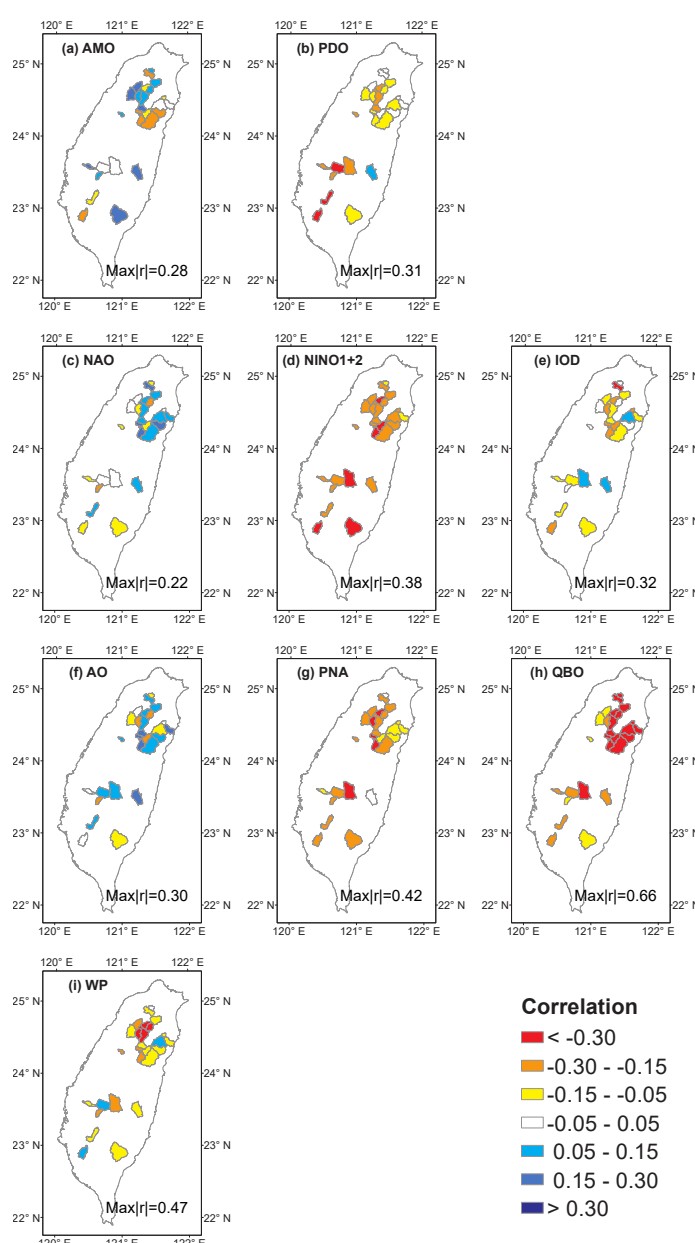

**Figure 3.** As in Figure 2, but for lagged correlations.





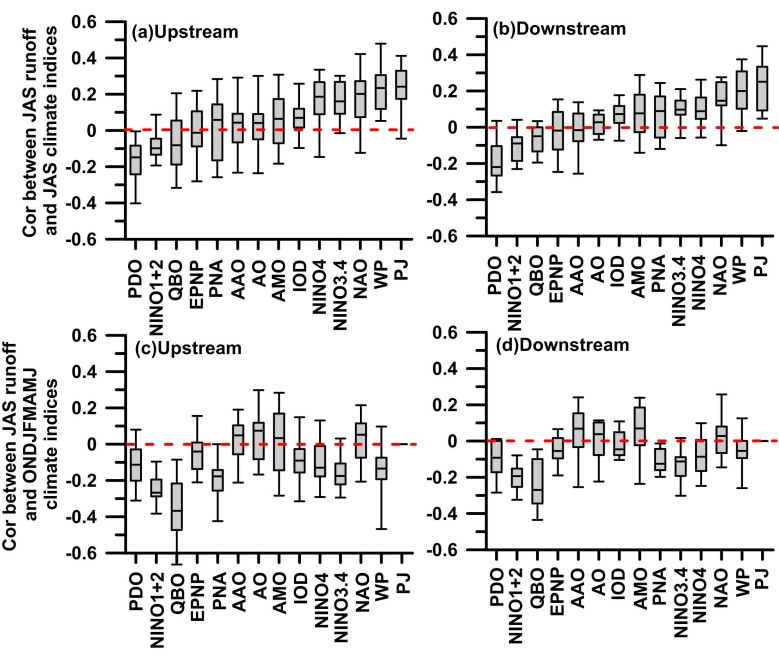

**Figure 4.** Box plots of correlation values of upstream (left) and downstream (right) runoff with climate indices.

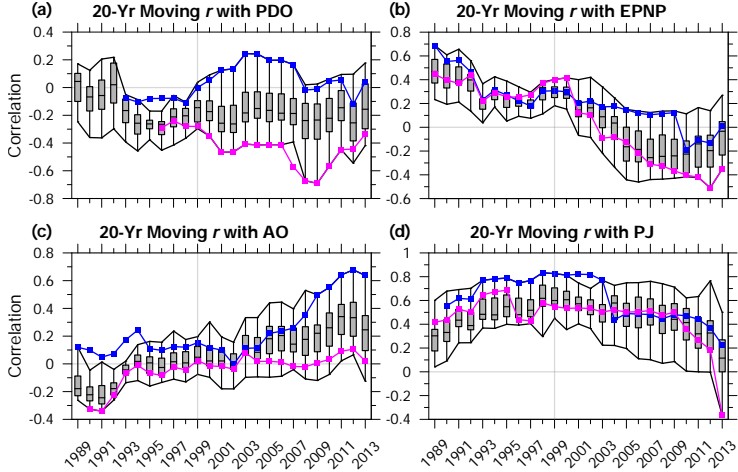

**Figure 5.** Selected moving-window correlation time series.





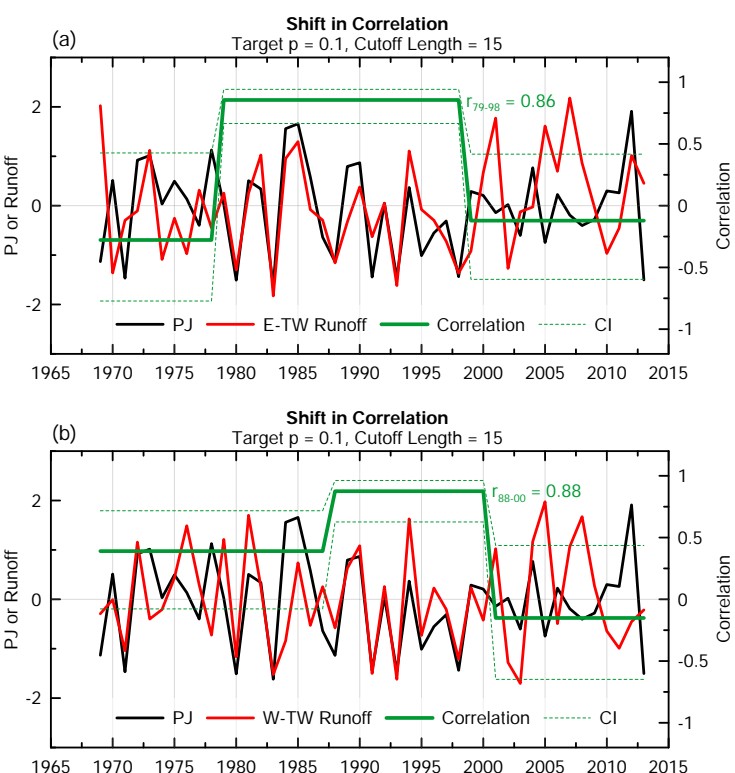

**Figure 6.** Shifts in the correlation between (top: East; bottom: West) Taiwan runoff and the PJ index.





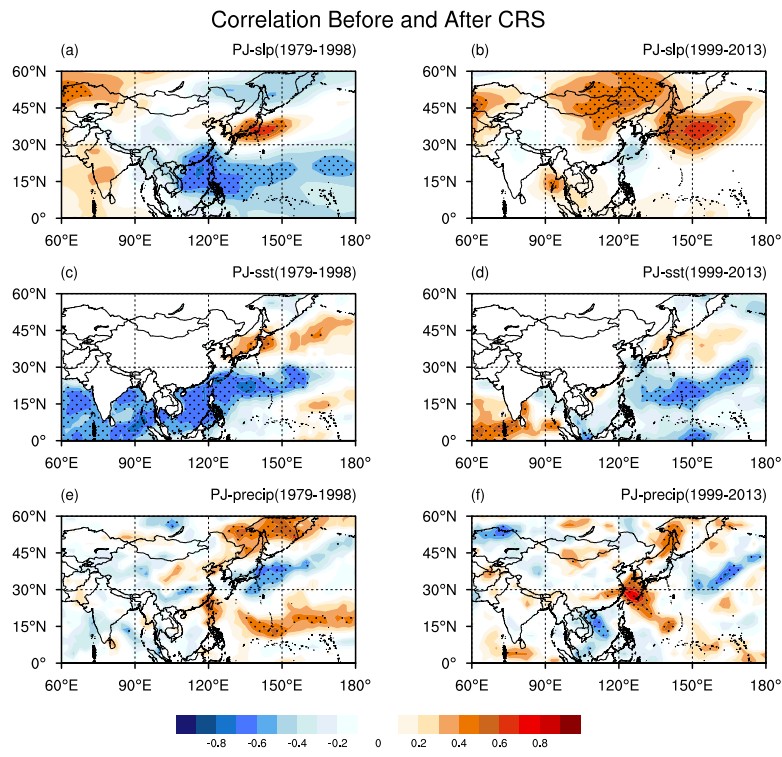

**Figure 7.** Correlation maps of the PJ index before (left panel) and after (right panel) year 1999. Top, center, and bottom panels are PJ vs. SLP, PJ vs. ERSST, and PJ vs. GPCP data, respectively. Correlation values at a 5% significance level are stippled.





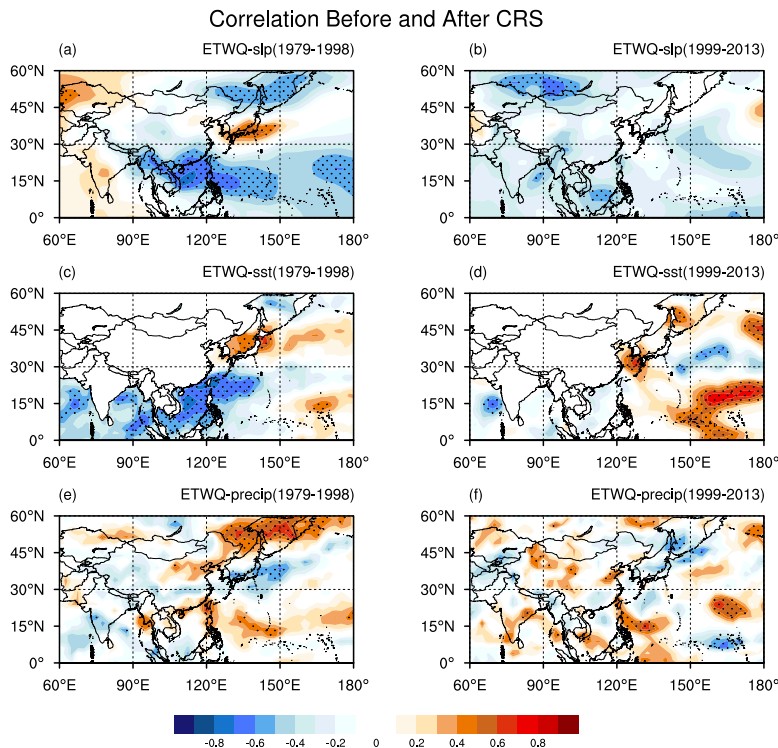

**Figure 8.** Correlation maps of the East Taiwan runoff (ETWQ) before (left panel) and after (right panel) year 1999. Top, center, and bottom panels are ETWQ vs. SLP, ETWQ vs. ERSST, and ETWQ vs. GPCP data, respectively. Correlation values at a 5% significance level are stippled.





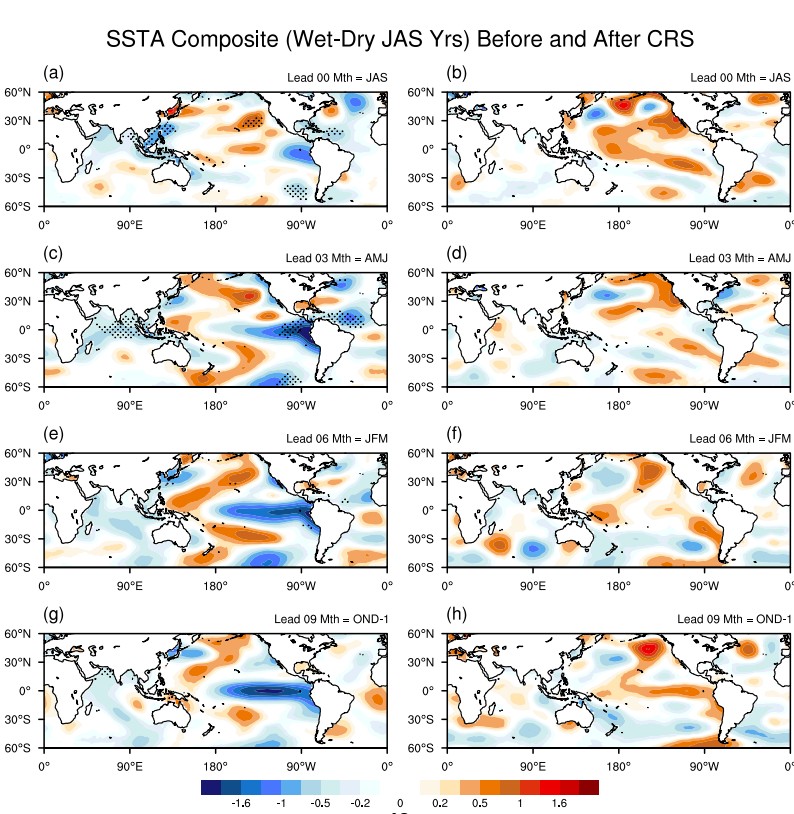

**Figure 9.** Evolution of SSTA composites from 0- to 9-month lead times based on wet-minus-dry years for the JAS ETWQ before (left panel) and after (right panel) year 1999. Stippled areas indicate the SSTA difference is at a 5% significance level according to Student's *t*-test.