# Peer review of "On the Relationship between Teleconnections and Taiwan's Streamflow: Evidence of Climate Regime Shift and Implications for Seasonal Forecasting"

_Hydrology and Earth System Sciences, 2016_

## Referee Comment (RC1) · Anonymous Referee #1 · 15 Jul 2016

This paper presents an analysis of correlations between large-scale climate indices and streamflow in 41 Taiwanese catchments. Additionally, a climate regime shift (CRS) analysis is employed to detect changes in the relationships between the climate indices and streamflow across time. Comments are made about the impact of CRS on predictor screening routines and forecasting.

The purpose of the paper is to identify the relationships between climate patterns and Taiwanese high season (July-August-September) streamflow. My understanding of the key findings suggested by the authors are: concurrent JAS correlations are positive

and high for West-Pacific, Pacific-Japan and NAO indices; 9-month averaged preceding climate indices (ONDJFMAMJ) are generally more weakly correlated with JAS streamflow with the exception of the QBO which is negatively and significantly correlated; and climate regime shifts occurred in the 1970s and 1990s.

I suspect the study will not vastly benefit the general seasonal streamflow forecasting community, however it could be of interest in the study region. The writing is not yet publication quality, there is not enough detail for the study to be repeatable, and some choices related to data prevent this study from being clear cut with robust conclusions. My overall opinion is the paper is not coherent enough to be published in HESS at this time. However, I do encourage the authors to rethink certain aspects of the study and seek eventual publication. My general and specific comments are below.

General comments

1) The stated purpose of this paper is to understand climate impacts on seasonal streamflow forecasting (as per the title) in Taiwan. Of concern is empirical prediction (P11 L5-6). P5 L22-23 states that lagged correlations are used to investigate forecasting possibilities. What is not made sufficiently clear is why the concurrent analysis of climate indices and streamflow is included in this study. To make use of concurrent relationships, models would need to be used to forecast the climate indices in the first instance. I suggest clarifying the reasoning and reconsider the weight given to the concurrent results in the paper unless knowing concurrent relationships is actually useful for empirical seasonal streamflow forecasting in Taiwan. Furthermore, the results and discussion interweave concurrent results and suggestions about the implications for forecasting in a way which I interpret as incompatible.

2) A different point, but related to the above. It severely bothers me that the Pacific Japan (PJ) correlations are different to all the others in that they are "semi-concurrent" (P5 L30-32) and not consistent with the separate concurrent and lagged analyses. The JJA PJ index is treated as concurrent for JAS, which in my mind is technically incorrect,

and concerning given the PJ results feature so heavily in the paper. The CRS analysis and discussion, as far as I can tell, hinges on the PJ index (Table 2 and P8 L24-27). It seems to me an effort has been made to include the PJ index because it yields high correlations (Fig 4) and garners some significant CRS results when really it should be excluded or included in a way that is consistent with the other results. Can the authors obtain the full PJ time series and complete the analysis more rigorously and put the PJ correlation and CRS analyses in the context of forecasting?

3) The manuscript is far too scant in the detail of the data and the methodology for it to be repeatable. Questions I have are: a) What is the period of data used? P4 L16 states that an "extended record" is selected. Exactly what is the period of data for each gauge? Are they all the same or different? b) What are the assumptions of the correlation analysis? What are its limitations and how is it suitable for this analysis? c) Is the CRS detection method robust for picking out multiple change points in short data records?

4) For me it would be far more interesting to analyse lagged correlations with increasing lead time as suggested at P5 L28-30, rather than averaging climate indices over the preceding three-, six- and nine-month periods and then presenting the best results. For forecasting, some of the indices are not immediately available after the end of June. E.g. at 15 July 2016, the PDO index for June is not yet published at jisao.washington.edu

Specific comments

5) Related to major comment 3, I suspect the QBO can have a strange distribution. Given it has the most outstanding lagged correlation (overall correlation actually) and the concluding remarks section trumpets its forecasting potential (P11 L10), this should be analysed further. It would be helpful to confirm that the results are not a quirk.

6) In the abstract and elsewhere it is mentioned that the lagged correlations represent 1 month lead time. From my experience, the lagged correlations would be called 0

months lead time, since data up until the end of June is used.

7) Should the RHS of equation (3) be 2(1-r)?

8) P6 L10 – not sure that it is correct to say x = y = 0 if the variables are normalized. The variance analysis doesn't appear to depend on this anyway.

9) Is it possible to mark significance thresholds on Figure 4? I understand it may not be possible if the data records have different lengths.

10) I may have missed the reasoning, but I don't understand why the pink and blue lines in Figure 5 start at different points

Technical corrections (typing errors, etc.)

11) Some captions are far too brief, e.g. Figure 5.

12) P7 L10: I suspect this should be *any* rather than *none*.

---

## Referee Comment (RC2) · Anonymous Referee #2 · 22 Aug 2016

General comments

The authors found significant correlation between Taiwan summer (July-September) catchment streamflow and the WP (West-Pacific) and PJ (Pacific-Japan) teleconnection indices, and demonstrated that the correlation relationship is not stable over a period of 50 years by calculating the correlation in a 20-year running window. Significantly high correlation appears only during the years from 1979-1999. The authors further used Rodionov's method to identify the correlation change points and found two significant points at 1988 and 2000. These findings prompted the authors to discuss

potential problems of using teleconnection indices as predictors for forecasting seasonal catchment streamflow. Although the subject is of great importance, the authors did not present sufficient scientific evidence to support the argument. I suggest the authors to continue the research and taking the following comments into account. The writing need to be more exact and concise.

Specific comments

1. P2L19-P2L26: I don't understand the point of this paragraph. "East Asia" is rather big compared with "Taiwan". Why is that "seeking the relationship between Taiwan's climate and large-scale circulations can provide some clue to direct the mechanisms of East Asian climate"? What are - the mechanisms of East Asian climate?

2. P3L14-P3L17: It seems to me that the authors wanted to use Taiwan as an example for "diagnosing underlying mechanisms of predictability and pointing caveats on intrinsic covariability between regional streamflow and large-scale circulation". However, the "predictability" and "caveats" depend strongly on the prediction model under discussion. Therefore, the "backbone" (P3L17) of the research should be a prediction model with acceptable prediction skill at least for a substantial period. In this regard, the following missing material is required to support the argument the authors trying to make.

(1) The authors need to present a prediction model that uses large-scale circulation to predict Taiwan streamflow with proved skill.

(2) The authors need to provide scientific evidence to explain the underlying mechanisms of the predictability of the prediction model.

(3) The authors need to show how variations of the intrinsic covariability influences the performance of the prediction model.

(4) With the evidence listed above, the authors can discuss the observed facts of climate regime shift and point out the caveats on using large-scale circulation indices to

predict regional streamflow.

3. P3L28-P3L31: The two objectives listed here cannot be "objectives" because the scientific questions/purposes are not clear.

4. P4-P6: The "Data and Analysis Procedures" section should focus on discussing the "Data" and "Procedure", such as the data length (e.g. beginning and end years, . . .), quality check (e.g. missing data issue and solution, . . .), methodology (e.g. decision principles of the teleconnection / large-scale indices, . . .). Other discussion such as the season (JAS) of study and references of teleconnection indices should be presented in the Introduction section.

5. P5L1-P5L14: The physical meaning of the teleconnection indices listed here is barely mentioned. It is not possible to diagnose "underlying mechanisms of predictability" without presenting the physical insight of the relationship between Taiwan climate and the large-scale circulation indices. Are all of the indices relevant to Taiwan climate variability?

6. The authors clearly showed that Taiwan JAS catchment streamflow is significantly correlated with the WP and PJ teleconnection indices and the correlation relationship changes with time. The large-scale climate also shows decadal-scale variations. However, time coincidence cannot be used for arguing physical relationship. For example, the argument of "The CRS firstly emanates from the change in the basin-scale climatology over the Pacific (e.g., shift in the PDO), and then the reorganized large-scale patterns can reset the relationship between the island-scale streamflow with established regional circulations (e.g., the PJ pattern)" on P9L12-14 is a hand-waving argument. Nothing is explained about how the decadal-scale changes of sea surfaces temperature (PDO) influences the regional circulation patter (PJ) in the atmosphere and then the rainfall pattern and subsequently the streamflow pattern in Taiwan.

There are many more hand-waving type of argument in the paper. There is no point to list out all of them here if the above points are not addressed.

---

## Author Comment (AC1) · 6 Sep 2016

**Interactive* comment on "On the Relationship between Teleconnections and Taiwan's Streamflow: Evidence of Climate Regime Shift and Implications for Seasonal Forecasting" *by* Chia-Jeng Chen and Tsung-Yu Lee**

**Anonymous Referee #1**

This paper presents an analysis of correlations between large-scale climate indices and streamflow in 41 Taiwanese catchments. Additionally, a climate regime shift (CRS) analysis is employed to detect changes in the relationships between the climate indices and streamflow across time. Comments are made about the impact of CRS on predictor screening routines and forecasting.

The purpose of the paper is to identify the relationships between climate patterns and Taiwanese high season (July-August-September) streamflow. My understanding of the key findings suggested by the authors are: concurrent JAS correlations are positive and high for West-Pacific, Pacific-Japan and NAO indices; 9-month averaged preceding climate indices (ONDJFMAMJ) are generally more weakly correlated with JAS streamflow with the exception of the QBO which is negatively and significantly correlated; and climate regime shifts occurred in the 1970s and 1990s.

I suspect the study will not vastly benefit the general seasonal streamflow forecasting community, however it could be of interest in the study region. The writing is not yet publication quality, there is not enough detail for the study to be repeatable, and some choices related to data prevent this study from being clear cut with robust conclusions. My overall opinion is the paper is not coherent enough to be published in HESS at this time. However, I do encourage the authors to rethink certain aspects of the study and seek eventual publication. My general and specific comments are below.

We thank the reviewer for making a considerable effort to review our manuscript and provide insightful comments. As addressed by the reviewer, we do believe that the results of our correlation analysis is of great importance to Taiwan and most East Asian regions sharing similar climatic conditions; however, we also believe that the discussion part of our study (i.e., effect of CRS on predictor screening in general) should deserve more attention in the forecasting community, especially for those applying empirical forecasting methods. To address your foremost concerns, we would like to make the following statements at the front of our point-by-point responses:

a)  Regarding the reproducibility of this study (details of the data and methodology used, in line with Comment 3):  We have listed all the periods of streamflow data used for the 41 gauges in Taiwan.  Assumptions and limitations of the correlation analysis will be amply discussed, and so will the robustness of the CRS detection method.  Related discussion can be found in our response to Comment 3 below.

b)  Regarding the inconsistent correlation analysis related to the Pacific-Japan (PJ) index (in line with Comment 2):  We have consulted with Dr. Hisayuki Kubota from the Japan Agency for Marine-Earth Science and Technology, who developed the PJ index in his journal paper, to obtain the raw pressure data at Yokohama and Hengchun for the derivation of the new PJ index in JAS.  The concurrent correlation analysis of the new PJ index is now consistent with all other indices, and our major findings stay very much the same, as expected.  Furthermore, we have also derived the PJ index in three preceding seasons (i.e., AMJ, JFM, and previous OND) for conducting more lagged correlation analyses (in line with Comment 4).  Please see our responses to Comments 2 and 4 below for more details.

We will carefully revise our article to your satisfactory level.  We wish our revision will find your support of our article.

General comments

1) The stated purpose of this paper is to understand climate impacts on seasonal streamflow forecasting (as per the title) in Taiwan.  Of concern is empirical prediction (P11 L5-6).  P5 L22-23 states that lagged correlations are used to investigate forecasting possibilities.  What is not made sufficiently clear is why the concurrent analysis of climate indices and streamflow is included in this study.  To make use of concurrent relationships, models would need to be used to forecast the climate indices in the first instance.  I suggest clarifying the reasoning and reconsider the weight given to the concurrent results in the paper unless knowing concurrent relationships is actually useful for empirical seasonal streamflow forecasting in Taiwan.  Furthermore, the results and discussion interweave concurrent results and suggestions about the implications for forecasting in a way which I interpret as incompatible.

We agree with the reviewer that the concurrent analysis does not produce immediate forecasting utility.  However, we believe that it is still important to examine concurrent relationships between climate indices and streamflow since many climate patterns have been proven to drive regional climates in the concurrent season.  Probably the idea of calculating contemporaneous/ concurrent correlations was best demonstrated by Wallace and Gutzler (1981), who nicely described several dominant teleconnection patterns at the Northern

Hemisphere extratropics during winter (e.g., NAO).  Beyond the Northern Hemisphere extratropics, one of the most important concurrent relationships being witnessed by several operating agencies and research organizations (e.g., CPC and IRI) is the impacts of ENSO on world regions.  Various maps of the concurrent relationships (e.g., composite and historical probability) have been archived as valuable references.  Over the Indian Ocean basin, the different phases of the IOD are also known to have pronounced concurrent impacts on the formation of the trade wind and the short rains over East Africa from October to November (Black et al., 2003; Clark et al., 2003; Behera et al., 2005; Chen and Georgakakos, 2015).  By contrast, significant lagged correlations (if identified) can indeed generate some forecasting utility, but to assess the dynamical mechanisms of the lagged relationships found by statistical approaches is usually not a trivial task.  To use concurrent relationships for forecasting, one can adopt a hierarchical or hybrid approach that applies another empirical or dynamical model to forecast the climate indices in the first instance (e.g., Kim and Webster, 2010; also suggested by the reviewer).

The above clarification will be incorporated into the revised article.

In addition, we wish to restate the original scope of this work, which was set to focus more on the discussion about the general relationships between teleconnections and Taiwan's streamflow, rather than the development of a prediction model for the pursuit of forecasting utility.  This article is currently included in the special issue of *sub-seasonal to seasonal hydrological forecasting* (as per the editor's suggestion), but we do not want to mislead the reviewer about our original intent.  Since we have agreed with the transfer decision of our manuscript, we certainly realize it should be our responsibility to include more forecasting elements in the study.  In essence, we have conducted more lagged correlation analyses as suggested by the reviewer in Comment 4 below.  This part of results will be included in the revised article to provide a more balanced weight between the concurrent and lagged results.

2) A different point, but related to the above.  It severely bothers me that the Pacific Japan (PJ) correlations are different to all the others in that they are "semi-concurrent" (P5 L30-32) and not consistent with the separate concurrent and lagged analyses.  The JJA PJ index is treated as concurrent for JAS, which in my mind is technically incorrect, and concerning given the PJ results feature so heavily in the paper.  The CRS analysis and discussion, as far as I can tell, hinges on the PJ index (Table 2 and P8 L24-27).  It seems to me an effort has been made to include the PJ index because it yields high correlations (Fig 4) and garners some significant CRS results when really it should be excluded or included in a way that is consistent with the other results.  Can the authors obtain the full PJ time series and complete the analysis more rigorously and put the PJ correlation and CRS analyses in the context of forecasting?

We think it would be worthwhile to provide the context of how we chose to use the PJ index. Kubota et al. (2015) developed the PJ index to monitor the long-term interannual variability of the PJ pattern (leading mode over the western North Pacific during summer), and they used JJA (typical definition of boreal summer) for the average period. It was unfortunate that they did not generate JAS or monthly PJ index for our purpose, so we ended up using their JJA index with an annotation (i.e., "semi-concurrent") added to the manuscript. Since the PJ pattern has been proven to dominate cyclonic activity and summer rainfall in East and Southeast Asia, it was our gut feeling that high correlations between the PJ index and Taiwan's streamflow should be observed as well. This supposition was then supported by our experiment, and then the intriguing CRS was identified. We believed the significant correlation and CRS results should outweigh the slight inconsistency with other indices, so we presented our findings as is without pursuing the absolute consistency.

Nevertheless, we agree that the PJ-related presentation is somewhat distinct from all other analyses, so we have managed to develop the new PJ index using the following steps (with Dr. Kubota's guidance):

a) We first obtained atmospheric pressure data at Yokohama and Hengchun in Japan and Taiwan, respectively.

b) We calculated the JAS (and all other tri-monthly periods, e.g., AMJ, JFM, and previous OND) average of atmospheric pressure anomaly, and then normalized the values by the standard deviation at each station.

c) The JAS (and all other tri-monthly periods) PJ index was derived from the difference of the two normalized pressure anomalies, and then the index was normalized again by its 1979–2009 standard deviation.

We have successfully reproduced the JJA PJ index and produced the new JAS PJ index. The figure below shows the comparison between the PJ index in JAS and that in JJA. The correlation analysis can now be performed with full consistency, and the results will be updated in the revised article (demonstrated by the table below). In terms of concurrent correlations, the updated results seem to be *even more significant* than the original ones.

[Figure]

**Figure R1.** Comparison between the PJ index in JAS and that in JJA.

**Table R1.** Comparison between the correlation values derived from JAS_PJ and those from JJA_PJ (original results in Table 2 of the manuscript); values before (after) the slash are concurrent (lagged) correlation coefficients ($\times 10^{-2}$, significant at $p = 0.05$ are bold and italic)

| JAS Runoff | Climate Index | |
|---|---|---|
| **Watershed** | **JAS_PJ** | **JJA_PJ†** |
| TC | *33*/-8 | *25*/* |
| HLO | *45*/21 | 32/* |
| WU | 26/12 | *45*/* |
| JS | *40/34* | *33*/* |
| BG | 13/8 | 26/* |
| ZW | 25/12 | 16/* |
| ER | 12/3 | 5/* |
| GP | *34*/3 | *30*/* |
| BN | *25*/-9 | 9/* |
| SGL | *40*/11 | *41*/* |
| HLI | *38*/13 | 20/* |
| HP | 17/-2 | 7/* |
| LY | *31*/-8 | 9/* |

†: lagged correlation was not computed for the original results

3) The manuscript is far too scant in the detail of the data and the methodology for it to be repeatable. Questions I have are: a) What is the period of data used? P4 L16 states that an "extended record" is selected. Exactly what is the period of data for each gauge? Are they all the same or different? b) What are the assumptions of the correlation analysis? What are its limitations and how is it suitable for this analysis? c) Is the CRS detection method robust for picking out multiple change points in short data records?

Please see our point-by-point responses to your specific questions:

a) Regarding the period of data:  The periods of record for all 41 catchments are listed in the table below (which will be included in the revised article). Despite the existence of CRS, correlation analysis typically requires a sufficiently long period of record.  Thus, we decided to use all available data even though their periods of record are not entirely the same.

**Table R2.** Period of data record and missing data percentage for all 41 catchments used for our analysis. Note that we use only JAS data in each year, and the missing data percentage is referred to as the percentage of years in which no JAS data is available.

| Catchment (downstream) | Period of Record | Missing Data % | Catchment (upstream) | Period of Record | Missing Data % | Catchment (upstream) | Period of Record | Missing Data % |
|---|---|---|---|---|---|---|---|---|
| TC | 1951–2013 | 0% | Cat_01 | 1970–2013 | 2.3% | Cat_15 | 1970–2013 | 2.3% |
| HLO | 1981–2013 | 0% | Cat_02 | 1970–2006 | 0% | Cat_16 | 1971–2013 | 0% |
| WU | 1966–2013 | 2.1% | Cat_03 | 1970–2002 | 0% | Cat_17 | 1970–2013 | 11.4% |
| JS | 1965–2009 | 0% | Cat_04 | 1970–2002 | 0% | Cat_18 | 1971–2013 | 2.3% |
| BG | 1949–2013 | 0% | Cat_05 | 1971–2007 | 0% | Cat_19 | 1970–2013 | 11.4% |
| ZW | 1960–2013 | 0% | Cat_06 | 1972–2013 | 2.4% | Cat_20 | 1970–2013 | 11.4% |
| ER | 1971–2013 | 0% | Cat_07 | 1970–2008 | 0% | Cat_21 | 1970–2013 | 11.4% |
| GP | 1951–2010 | 0% | Cat_08 | 1976–2013 | 5.3% | Cat_22 | 1970–2001 | 0% |
| BN | 1948–2013 | 4.5% | Cat_09 | 1972–2013 | 0% | Cat_23 | 1974–2013 | 5% |
| SGL | 1969–2013 | 0% | Cat_10 | 1970–2013 | 0% | Cat_24 | 1977–2011 | 5.7% |
| HLI | 1969–2013 | 0% | Cat_11 | 1970–2013 | 2.3% | Cat_25 | 1977–2011 | 2.9% |
| HP | 1975–2013 | 5.1% | Cat_12 | 1970–2008 | 0% | Cat_26 | 1970–2012 | 2.3% |
| LY | 1949–2009 | 0% | Cat_13 | 1970–2013 | 9.1% | Cat_27 | 1970–2013 | 0% |
| | | | Cat_14 | 1970–2013 | 4.5% | Cat_28 | 1970–2013 | 2.3% |

b) Regarding the assumptions and limitations of the correlation analysis: One of the most fundamental assumptions of the correlation analysis would be the result of such analysis does not indicate any causality.  The result can be two-way; that is, there is no physical implication for a predictor-predictand relationship.  However, the assumption taken by us is that significant correlations should suggest some large-scale dominance over local-scale hydroclimate since the opposite route of dominance (i.e., the impact of a disturbance at the island (Taiwan) scale on large-scale circulations) is unlikely and hard to explain.  In addition, we also neglect the effect of any outliers (if exist) and examine only the linear relationship between two continuous variables.  In other words, our analysis cannot identify any nonlinear effect of extreme teleconnection patterns on Taiwan's streamflow.

c) Regarding any caveats of using the CRS detection method (number of change points vs. data length):  As already indicated by the manuscript, there are several classes of approaches available for the detection of regime shifts, such as parametric methods (e.g., the classical $t$-test), non-parametric methods (e.g., the Mann-Whitney-Pettitt test), regression-based methods, cumulative sum methods, and sequential methods.  Rodionov (2005) pointed out the pros and cons of some common approaches as well as his sequential method (Rodionov, 2004).  The pros of his method are the automatic, early detection of a regime shift and the ability to monitor a possibility of a regime shift in real time.  He has shown that his method can outperform Lanzante's method (another robust, non-parametric procedure developed by Lanzante, 1996).  Therefore, since developed, Rodionov's method has been used in many studies in climate sciences.  Nevertheless, the cons of his method are the requirement of some experimentation on the two parameters used (i.e., the cut-off length $l$ and probability level $p$) and inability to account for a gradual regime shift and data with obvious autocorrelation (or red noise, but this issue was later ameliorated by a prewhitening procedure introduced by him, Rodionov, 2006).  According to Rodionov (2004), while the probability level $p$ is known to determine the critical value of $t$ (the Student's $t$-distribution), *"the cut-off length $l$ determines the minimum length of the regimes, for which the magnitude of the shifts remains intact."*  It has been tested that, the larger $l$ is set, the fewer change points can be identified.  By contrast, a smaller $l$ does not necessarily lead to more change points since only those significant change points can be identified based on the $t$-test.  In other words, if there is no strong regime shift in the data series, the method with some variations in $p$ and $l$ simply cannot identify any change point (Rodionov, 2015 and verified by us too).  In any case, we should still use Rodionov's method with caution since the CRS detection method is purely statistical, and the CRS results could be meaningless without solid evidence from related research.

We will incorporate the above discussions into the revised article.

4) For me it would be far more interesting to analyse lagged correlations with increasing lead time as suggested at P5 L28-30, rather than averaging climate indices over the preceding three-, six- and nine-month periods and then presenting the best results.  For forecasting, some of the indices are not immediately available after the end of June. E.g. at 15 July 2016, the PDO index for June is not yet published at jisao.washington.edu

We have been aware of the reviewer's (and likely other readers') interest in seeing more results of lagged correlations.  Thus, for all 41 catchments, we have calculated two additional sets of lagged correlations between the JAS flow data

and the climate indices averaged over two preceding periods (ONDJFM and OND), in line with our average scheme used in the article.  The table below shows the new generated results of lagged correlations for the major watersheds in Taiwan; the complete results will be included in the revised article.

**Table R3.** Results of correlation analysis for the major watersheds in Taiwan; values before (after) the slash are lagged correlation coefficients with preceding ONDJFM (OND) climate indices ($\times 10^{-2}$, significant at $p = 0.05$ are bold and italic).

| JAS Runoff Watershed | Climate Index | | | | | | | | | | | | | |
|---|---|---|---|---|---|---|---|---|---|---|---|---|---|---|
| | AMO | PDO | NINO1+2 | NINO3.4 | NINO4 | IOD | EPNP | PNA | AO | AAO | NAO | QBO | WP | PJ |
| TC | 4/5 | 4/7 | -13/-14 | -13/-14 | -1/-1 | -10/-14 | -1/12 | -9/-9 | -6/-1 | 12/0 | 1/6 | -20/-19 | -14/-9 | -11/-2 |
| HLO | 27/29 | -14/-10 | -32/-29 | -24/-23 | -12/-10 | -3/-5 | -27/-10 | -11/0 | -5/4 | 4/-15 | 2/12 | ***-42/-43*** | -4/0 | 21/20 |
| WU | -22/-19 | 0/7 | -18/-17 | -14/-17 | -11/-18 | -14/-24 | -4/1 | -11/-2 | 7/8 | 6/-6 | 14/14 | -1/3 | -4/-3 | 10/4 |
| JS | 8/15 | -16/-9 | -28/***-29*** | ***-32/-31*** | -26/-26 | -13/-19 | -26/-11 | -15/8 | 8/12 | ***30***/8 | 8/21 | -21/-17 | 9/6 | ***43/34*** |
| BG | -6/-7 | 5/12 | -12/-12 | -12/-10 | -18/-18 | -7/-14 | -2/18 | 2/9 | ***-28***/-13 | -9/-21 | -4/12 | -20/-17 | -1/-10 | 1/16 |
| ZW | 4/8 | -26/-19 | -25/***-28*** | -22/-21 | -19/-17 | -12/-14 | -20/-8 | -16/-3 | 4/2 | 12/-11 | -5/3 | ***-28***/-26 | 0/-9 | 21/23 |
| ER | 16/18 | -10/-7 | -16/-10 | -15/-13 | -11/-8 | -7/-13 | -14/0 | -5/0 | -1/-7 | 22/5 | 0/0 | 3/9 | 8/4 | 21/21 |
| GP | 1/6 | -9/0 | -20/-21 | -21/-19 | -17/-17 | -9/-19 | -4/10 | -13/-5 | 5/2 | ***28***/6 | 3/9 | -20/-17 | 10/5 | 13/19 |
| BN | 16/***25*** | -22/-24 | ***-25/-26*** | -15/-17 | -9/-12 | 8/5 | 2/-9 | -15/-20 | -13/-11 | 24/5 | 3/-7 | -9/-7 | -11/-15 | -7/-5 |
| SGL | -5/-3 | 4/5 | -16/-7 | -8/8 | -2/-1 | -4/2 | -11/-5 | -11/-8 | 0/-2 | 21/16 | 6/8 | ***-31***/-24 | -8/-20 | 9/7 |
| HLI | 21/21 | -17/-16 | -22/-18 | -9/-7 | 1/2 | -2/0 | -26/-14 | -15/-3 | -12/-8 | 17/0 | -12/3 | ***-34/-29*** | 8/-13 | 20/9 |
| HP | 13/14 | 1/8 | -7/-4 | 3/5 | 12/13 | 7/9 | -1/13 | -20/-23 | 5/-1 | -6/-10 | -6/-1 | -28/-22 | -13/-30 | 11/1 |
| LY | 18/18 | -5/-5 | -11/-15 | -12/-14 | -5/-3 | 6/-5 | 7/6 | -4/-7 | -17/-17 | -1/-11 | 6/-5 | ***-33/-25*** | 1/-2 | 4/8 |

Specific comments

5) Related to major comment 3, I suspect the QBO can have a strange distribution. Given it has the most outstanding lagged correlation (overall correlation actually) and the concluding remarks section trumpets its forecasting potential (P11 L10), this should be analysed further. It would be helpful to confirm that the results are not a quirk.

First of all, the QBO depicts a quasi-periodic oscillation between easterlies (positive) and westerlies (negative phase) over the lower tropical stratosphere, and the period is about 20 to 36 months. This information, as per the other reviewer's suggestion, will be added to Section 2.1 in the revised article.

In order to confirm that the most outstanding lagged correlation between Taiwan's streamflow and the QBO, additional literature review and field significance test are conducted. The strongest lagged correlation is very likely attributed to the tropical cyclone (TC) activity in the western North Pacific (WNP) modulated by the QBO. Chan (1995) has performed a cross-spectral analysis between the QBO and the number of TCs in the WNP and indicated that the leading westerly phase of the QBO can result in an increase in TC activity. He explained that the westerly phase of the QBO creates an environment of relatively low vertical wind shear in favor of TC formation. Ho et al. (2009) later found that during the westerly (easterly) phase of the QBO, more TCs approaches the East China Sea (the eastern shore of Japan). Therefore, the negative correlation between the QBO index and TC activity in the vicinity of Taiwan is carried over into the negative correlation with streamflow. In fact, such strong correlations found in 22 out of the 41 catchments also reach field significance. The number of catchments with significant temporal correlations has exceeded the critical value of field significance ($p = 0.05$) from the empirical null distribution (Figure R2) developed by using a Monte Carlo technique similar to those suggested by Livezey and Chen (1983) and Wilks (2011). 2000 Monte Carlo trials are used, and each trial depicts a significant local test for correlations between the "randomly ordered" QBO index and streamflow data at the 41 catchments, resulting a count of the number of catchments with significant temporal correlations constituting the null distribution.

[Figure]

**Figure R2.** Empirical null distribution of field significance test. The abscissa represents the number of catchments with correlations between streamflow and lagged QBO significant at the 95% level ($p = 0.05$) in 2000 Monte Carlo trials.

6) In the abstract and elsewhere it is mentioned that the lagged correlations represent 1 month lead time. From my experience, the lagged correlations would be called 0 months lead time, since data up until the end of June is used.

You are correct, but to avoid any confusion with the concurrent analysis, we will replace those lead time information with the exact average period. For example, we will make the following change in the abstract (original manuscript P1L8):

*"On the other hand, the Quasi-Biennial Oscillation index averaged over a period from previous October to concurrent June significantly correlate with the JAS flows (most significant r = -0.66), indicating some forecasting utility."*

7) Should the RHS of equation (3) be 2(1-r)?

Sorry for the typo, and yes, it should be 2(1-r).

8) P6 L10 – not sure that it is correct to say x = y = 0 if the variables are normalized. The variance analysis doesn't appear to depend on this anyway.

We agree that "normalization" is a term with a more rigorous definition, so we will revise the sentence as:

*"Further, if the two variables have zero mean and unit variance, the above equation…"*

9) Is it possible to mark significance thresholds on Figure 4?  I understand it may not be possible if the data records have different lengths.

As indicated by the reviewer, since the data records have different lengths, it is not possible to mark significance thresholds on Figure 4.

10) I may have missed the reasoning, but I don't understand why the pink and blue lines in Figure 5 start at different points.

Same reason above, since the data records have different lengths and start at different years, the pink and blue lines in Figure 5 start at different points.  We will make this remark in the revised article.

Technical corrections (typing errors, etc.)

11) Some captions are far too brief, e.g. Figure 5.

Agreed.  We will add more descriptions to those short captions.  For instance, the caption of Figure 5 will be revised as:

> *"**Figure 5.** Selected moving-window correlation results. Each boxplot encapsulates correlation values derived from 28 upstream catchments (JAS runoff vs. specific climate index). Blue (magenta) time series denotes the highest (lowest) moving-window correlations over the temporal horizon. Please refer to Section 3.2 for more details."*

12) P7 L10: I suspect this should be \*any\* rather than \*none\*.

Agreed.  We will change to "any."

**References**

Behera SK, Luo JJ, Masson S, Delecluse P, Gualdi S, Navarra A, Yamagata T. 2005. Paramount impact of the Indian Ocean dipole on the East African short rains: A CGCM study. J. Clim. 18: 4514–4530.

Black E, Slingo J, Sperber KR. 2003. An observational study of the relationship between excessively strong short rains in coastal East Africa and Indian Ocean SST. Mon. Weather Rev. 131: 74–94.

Chan, J.C., 1995. Tropical cyclone activity in the western North Pacific in relation to the stratospheric quasi-biennial oscillation. Monthly Weather Review, 123(8), pp.2567-2571.

Clark CO, Webster PJ, Cole JE. 2003. Interdecadal variability of the relationship between the Indian Ocean zonal mode and East African coastal rainfall anomalies. J. Clim. 16: 548–554.

Ho, C.H., Kim, H.S., Jeong, J.H. and Son, S.W., 2009. Influence of stratospheric quasi-biennial oscillation on tropical cyclone tracks in the western North Pacific. Geophysical Research Letters, 36(6).

Kim, H.M. and Webster, P.J., 2010. Extended-range seasonal hurricane forecasts for the North Atlantic with a hybrid dynamical-statistical model. Geophysical Research Letters, 37(21).

Lanzante, J.R., 1996. Resistant, robust and non-parametric techniques for the analysis of climate data: Theory and examples, including applications to historical radiosonde station data. International Journal of Climatology, 16(11), pp.1197-1226.

Livezey, R.E. and Chen, W.Y., 1983. Statistical field significance and its determination by Monte Carlo techniques. Monthly Weather Review, 111(1), pp.46-59.

Rodionov, S.N., 2005. A brief overview of the regime shift detection methods. Large-Scale Disturbances (Regime Shifts) and Recovery in Aquatic Ecosystems: Challenges for Management Toward Sustainability, pp.17-24.

Rodionov, S.N., 2006. Use of prewhitening in climate regime shift detection. Geophysical Research Letters, 33(12).

Wilks, D.S., 2011. Statistical methods in the atmospheric sciences (Vol. 100). Academic press.

---

## Author Comment (AC2) · 6 Sep 2016

**Interactive* comment on "On the Relationship between Teleconnections and Taiwan's Streamflow: Evidence of Climate Regime Shift and Implications for Seasonal Forecasting" *by* Chia-Jeng Chen and Tsung-Yu Lee**

**Anonymous Referee #2**

General comments

The authors found significant correlation between Taiwan summer (July-September) catchment streamflow and the WP (West-Pacific) and PJ (Pacific-Japan) teleconnection indices, and demonstrated that the correlation relationship is not stable over a period of 50 years by calculating the correlation in a 20-year running window. Significantly high correlation appears only during the years from 1979–1999. The authors further used Rodionov's method to identify the correlation change points and found two significant points at 1988 and 2000. These findings prompted the authors to discuss potential problems of using teleconnection indices as predictors for forecasting seasonal catchment streamflow. Although the subject is of great importance, the authors did not present sufficient scientific evidence to support the argument. I suggest the authors to continue the research and taking the following comments into account. The writing need to be more exact and concise.

We are grateful for the reviewer's insightful comments (special thanks to the notice of the importance of the subject), which we address in detail below. We will incorporate our responses to your comments into our revision, and look forward to the re-evaluation of the article for publication.

Specific comments

1) P2L19-P2L26: I don't understand the point of this paragraph. "East Asia" is rather big compared with "Taiwan." Why is that "seeking the relationship between Taiwan's climate and large-scale circulations can provide some clue to dissect the mechanisms of East Asian climate?" What are the mechanisms of East Asian climate?

In that paragraph, we merely wanted to use the analogy between the weather systems found in Taiwan and those in East Asia to indicate our findings could be applicable to other East Asian regions. Climate similarity among these areas (e.g., Taiwan and south-to-southeast China) can be identified by employing EOF analysis (e.g., Wu et al., 2009). Nevertheless, we acknowledge that this

paragraph might not be precise enough to clearly depict our motivation, and the scale difference between Taiwan and East Asia could be somewhat confusing. As noted by the reviewer too (e.g., next comment), the scope of this study is certainly *not* to scrutinize the mechanisms of East Asian climate. In accordance with your fourth comment below, in the revised article, we will rearrange the two paragraphs in Sections 1 and 2 to compose a new paragraph to emphasize more Taiwan's climates and our motivation. The new paragraph will be as follows:

> *"Several studies (e.g., Wang et al., 2000; Yang et al., 2002; Wang and Fan, 2005; Choi et al., 2012) have witnessed the various effects of teleconnection patterns on East Asian regions, in which an island country Taiwan is situated (Figure 1). Taiwan has an area about 36,000 km² and features most weather systems found in East Asia, including spring rains, Mei-Yu, and East Asian monsoon from spring to summer, typhoons from summer to autumn, and the Mongolian high pressure system and associated northeast monsoon in winter. Because of the Central Mountain Range (topographic variations) and gradually varied climate zones (latitudinal differences), the influence of those weather systems on precipitation in particular can show great east-west and north-south contrasts. As a result, while the wet season generally spans from summer to autumn based on the long-term average, Taiwan's precipitation and streamflow in the wet season exhibits great spatial distributions of prominent intra-seasonal and inter-annual variations. Thus, seeking the relationship between Taiwan's climate in the wet season and large-scale circulations can guide the development of a hydro-climatic forecasting framework potentially of benefit to water resources management in this area."*

2) P3L14-P3L17: It seems to me that the authors wanted to use Taiwan as an example for "diagnosing underlying mechanisms of predictability and pointing caveats on intrinsic covariability between regional streamflow and large-scale circulation." However, the "predictability" and "caveats" depend strongly on the prediction model under discussion. Therefore, the "backbone" (P3L17) of the research should be a prediction model with acceptable prediction skill at least for a substantial period. In this regard, the following missing material is required to support the argument the authors trying to make.

2-1) The authors need to present a prediction model that uses large-scale circulation to predict Taiwan streamflow with proved skill.

2-2) The authors need to provide scientific evidence to explain the underlying mechanisms of the predictability of the prediction model.

2-3) The authors need to show how variations of the intrinsic covariability influences the performance of the prediction model.

2-4) With the evidence listed above, the authors can discuss the observed facts of climate regime shift and point out the caveats on using large-scale circulation indices to predict regional streamflow.

We totally agree that the inclusion of a prediction model with acceptable skill in our study can facilitate the discussion about how regional streamflow prediction can be affected by potential climate regime shifts (CRS). In fact, it is our intention to develop a new hydro-climatic forecasting method that takes the effect of CRS into account. The new forecasting method is going to be a renovation of the "dipole forecasting model," previously developed by the coauthor and his colleague (Chen and Georgakakos, 2014) and designed to excel at auto-identifying dipole-like predictor patterns from various oceanic and atmospheric fields (e.g., SST, SLP, geopotential height, and wind). However, we are still in the process of brainstorming the most adequate plug-in module of the CRS detection for the dipole forecasting model. Even though this work is still ongoing, we try to incorporate the reviewer's suggestion into our article by performing multiple linear regression as a surrogate for our new forecasting method (yet to come) to illustrate the effect of CRS on streamflow prediction in this study.

Linear regression is widely used in climate forecasting studies (e.g., Hastenrath 1995, 2004; Chen and Georgakakos, 2015) to generate forecasts based on a calibrated, linear equation that depicts how a hydro-climatic predictand ($\mathbf{Y}$, streamflow in our case) responds to selected predictors ($\mathbf{X}$, climate indices in our case):

$$\mathbf{Y} = \mathbf{X}\beta, \tag{R1}$$

where $\beta$ are coefficients estimated by ordinary least squares. In each catchment, we will develop a linear regression equation as the prediction model. In terms of predictors (i.e., independent variables), we adopt 13 climate indices described in the paper (with AAO excluded as relatively short in record) and perform stepwise model selection based AIC (Akaike Information Criteria). Model selection can be performed in forward or backward direction, and we use both directions to ensure a thorough search in the variable space. Afterwards, to avoid possible multicollinearity issues resulting from some highly correlated climate indices, the variance inflation factor (VIF) is assessed:

$$VIF_j = \frac{1}{1 - R_j^2}, \tag{R2}$$

where $R_j^2$ is the coefficient of determination from a regression of the $j^{th}$ predictor on any other predictors. According to the literature (Chen and Georgakakos, 2014; Hidalgo-Muñoz et al., 2015), the VIF tolerance threshold is set to be 4 for small samples (say ~50 points). The final model is thus determined and used for generating hindcasts (i.e., retrospective forecasts) for that catchment. The generation of hindcasts is subject to the leave-one-out cross-validation (LOOCV) procedure to circumvent artificial skill. Eventually, the LOOCV correlation and Gerrity Skill Score (GSS, Gerrity, 1992) are calculated to assess the prediction

skill in that catchment.  The above framework will be repeated for all 41 catchments.

Figure R1 below shows some hindcasting results for selected upstream and downstream catchments.  LOOCV correlations vary from one catchment to another, and can be as high as ~*0.6*.  As a more stringent metric, cross-validated GSS values are generally lower, but most of time pass the significant threshold (e.g., ~*0.25* for data size 30, determined by the bootstrap-based hypothesis testing, Chen and Georgakakos, 2014).  Overall, using large-scale circulation indices can produce fair to good prediction skills in summer streamflow prediction in Taiwan.  Among those many climate indices, the PDO and PJ indices are selected most frequently as the predictors for the catchments [while the former is selected seven times (except for SGL), the latter is selected five times (except for Catchments 3 and 9 and BG) for the results shown in Figure R1].  This result is to a certain extent consistent with our general correlation assessments (e.g., Table 2 in our original manuscript) and indicates the general dominance of summer climate in Taiwan.  Regarding the origin of the predictability, Chen and Chen (2011) indicated that the PDO coincides with the specific meridional SST contrast (i.e., warming in the tropical central and eastern Pacific and cooling in the extratropical North Pacific), which plays a dominant role in modulating summer rainfall in Taiwan.  Choi et al. (2010), Kosaka et al. (2013), and Kubota et al. (2016) all provided sufficient evidence of the significant impact of the PJ on tropical cyclone activity and rainfall over the western North Pacific during summer.  Based on our findings, the predictability for summer rainfall can be extended to streamflow in Taiwan.

From Figure R1, we can note that the relatively better performance of each LOOCV time series occurs during the period from the late 1970s to the late 1990s, which coincides with the CRS epoch discussed in the manuscript.  Hindcasts during the pre-regime shift epoch seem to be still able to capture the general variability of observed runoff (with relatively poorer performance), whereas hindcasts during the post-regime shift epoch appear to present more opposite signals and apparent departures from observed runoff.  It is worth noting that some of the departures occur in years when JAS typhoon activity is abnormally high.  For example, in 2007, typhoons Pabuk, Sepat, and Wipha together generated the highest amount of cumulative rainfall for some watersheds over the past decade, and in 2008, typhoons Kalmaegi, Fung-Wong, Sinlaku, Hagupit, and Jangmi made a record of continuous invasions of intense typhoons (all Category 2 and above) in JAS.  To further illustrate the effect of CRS on streamflow prediction, we fit a new regression model using the data from 1979 to 1998, and then evaluate how the fitted model performs in the remaining years.  Using the SGL watershed as an example, Figure R2 shows the new hindcasting result.  In comparison with the bottom-left plot in Figure R1, the new fitted model exhibits some definite improvement during the period from 1979 to 1998, showing the outstanding CV correlation and GSS values as *0.84* and *0.56*, respectively.  However, the fitted model can generate nothing but extremely poor

hindcasts for the remainders.  In fact, both skill metrics show a reverse sign, clearly illustrating distinct climate regimes over the temporal horizon.  In contrast to the above experiment, if we fit another regression model using the data outside the time frame of 1979–98, the stepwise model selection scheme discloses *no* climate indices qualified for being a predictor.  To sum up, the linear regression experiment points out that the assumption of a stable predictor-predictand relationship could be quite problematic for hydro-climatic forecasting due to the observation of CRS.

In addition to the above response to your specific comment, we would like to forward the following message (response to the other reviewer) to you since we wonder if the inclusion of our article in the special issue has caused any misconceptions about evaluating our work:

> *"We wish to restate the original scope of this work, which was set to focus more on the discussion about the general relationships between teleconnections and Taiwan's streamflow, rather than the development of a prediction model for the pursuit of forecasting utility.  This article is currently included in the special issue of 'sub-seasonal to seasonal hydrological forecasting' (as per the editor's suggestion), but we do not want to mislead the reviewer about our original intent.  Since we have agreed with the transfer decision of our manuscript, we certainly realize it should be our responsibility to include more forecasting elements in the study."*

In essence, as indicated above, we have developed a prediction model using linear regression to support our argument.  This part of analysis and associated results will be included in the revised article (likely as new Section 3.3) to make our work fit better within the scope of the special issue.

[Figure]

**Figure R1.** Selected hindcasting results for upstream and downstream catchments in Taiwan using linear regression. Time series in red are model estimates based on the leave-one-out cross-validation (LOOCV) procedure. Cross-validated (CV) correlation and GSS values are also denoted in each plot.

[Figure]

**Figure R2.** Similar to the bottom-left plot in Figure R1, but the linear regression model is trained/fitted with the data from 1979 to 1998, and then the fitted model is tested with the rest of data points (i.e., 1969–78 and 1999–2013).

3) P3L28-P3L31:  The two objectives listed here cannot be "objectives" because the scientific questions/purposes are not clear.

We wish our responses to the above comments (as well as other comments below) have made the scientific questions/purpose clear.  Moreover, another specific purpose as already stated in the paragraph above (original manuscript P3L3–17) is that a similar analysis has not conducted before in this area to the best of our knowledge.

We will revise the second objective as:
> *"To verify the existence of any CRS signals in the correlation and to discuss associated changes in large-scale circulation patterns."*

We will also add a third objective:
> *"To illustrate the overall prediction skill and the effect of CRS on streamflow prediction in Taiwan using a linear regression approach."*

4) P4-P6:  The "Data and Analysis Procedures" section should focus on discussing the "Data" and "Procedure", such as the data length (e.g. beginning and end years), quality check (e.g. missing data issue and solution), and methodology (e.g. decision principles of the teleconnection/large-scale indices). Other discussion such as the season (JAS) of study and references of teleconnection indices should be presented in the Introduction section.

As pointed out by the first reviewer as well, the information about data length and quality check will be amply supplemented in the revised article.  We also list the periods of record and missing data percentage for all 41 catchments in the table below.  Note that the periods of record are not entirely the same, and we decided to use all available data for the calculation of correlation values.  30 out of 41 gauges present missing data less than 3% (e.g., 1 out of 40 years is missing), indicating the quality of JAS flow data is quite reasonable.  For those missing-data years, we do not perform any data filling because we do not want to create any artificial, subjective flow quantities that may skew correlation values; that is, we simply skip the pair of data (flow and climate index) in those missing-data years for the calculation of correlation values.

**Table R1.** Period of data record and missing data percentage for all 41 catchments used for our analysis. Note that we use only JAS data in each year, and the missing data percentage is referred to as the percentage of years in which no JAS data is available.

| Catchment (downstream) | Period of Record | Missing Data % | Catchment (upstream) | Period of Record | Missing Data % | Catchment (upstream) | Period of Record | Missing Data % |
|---|---|---|---|---|---|---|---|---|
| TC | 1951–2013 | 0% | Cat_01 | 1970–2013 | 2.3% | Cat_15 | 1970–2013 | 2.3% |
| HLO | 1981–2013 | 0% | Cat_02 | 1970–2006 | 0% | Cat_16 | 1971–2013 | 0% |
| WU | 1966–2013 | 2.1% | Cat_03 | 1970–2002 | 0% | Cat_17 | 1970–2013 | 11.4% |
| JS | 1965–2009 | 0% | Cat_04 | 1970–2002 | 0% | Cat_18 | 1971–2013 | 2.3% |
| BG | 1949–2013 | 0% | Cat_05 | 1971–2007 | 0% | Cat_19 | 1970–2013 | 11.4% |
| ZW | 1960–2013 | 0% | Cat_06 | 1972–2013 | 2.4% | Cat_20 | 1970–2013 | 11.4% |
| ER | 1971–2013 | 0% | Cat_07 | 1970–2008 | 0% | Cat_21 | 1970–2013 | 11.4% |
| GP | 1951–2010 | 0% | Cat_08 | 1976–2013 | 5.3% | Cat_22 | 1970–2001 | 0% |
| BN | 1948–2013 | 4.5% | Cat_09 | 1972–2013 | 0% | Cat_23 | 1974–2013 | 5% |
| SGL | 1969–2013 | 0% | Cat_10 | 1970–2013 | 0% | Cat_24 | 1977–2011 | 5.7% |
| HLI | 1969–2013 | 0% | Cat_11 | 1970–2013 | 2.3% | Cat_25 | 1977–2011 | 2.9% |
| HP | 1975–2013 | 5.1% | Cat_12 | 1970–2008 | 0% | Cat_26 | 1970–2012 | 2.3% |
| LY | 1949–2009 | 0% | Cat_13 | 1970–2013 | 9.1% | Cat_27 | 1970–2013 | 0% |
| | | | Cat_14 | 1970–2013 | 4.5% | Cat_28 | 1970–2013 | 2.3% |

As aforementioned, we will move some of the general description of Taiwan, its climates (P4L3–8), and the target high-flow season (P4L22–25) to the introduction section and merge them with the revised third paragraph in response to your first comment.

Regarding the references of teleconnection indices, most of them have already been presented in the first two paragraphs of the original manuscript. The paragraph in Section 2.1 mainly focuses on explaining the decision principles of the teleconnection indices, as pointed out by the reviewer. In accordance with the comment below, we will add some short descriptions of the physical meaning of the teleconnection indices in that paragraph.

5) P5L1-P5L14: The physical meaning of the teleconnection indices listed here is barely mentioned. It is not possible to diagnose "underlying mechanisms of predictability" without presenting the physical insight of the relationship between Taiwan climate and the large-scale circulation indices. Are all of the indices relevant to Taiwan climate variability?

Many of the teleconnection indices (e.g., ENSO, NAO, and PDO) are well known and their physical meaning have been substantially elaborated in dedicated articles (e.g., Trenberth, 1997; Trenberth and Stepaniak, 2001; Hurrell, 1995; Mantua et al., 1997). However, we support the reviewer's assertion that we should at least summarize the physical definition of each teleconnection index in that paragraph.

For instance, the ENSO is characterized as an air-sea coupled phenomenon: a zonal Sea Level Pressure (SLP) anomaly in the tropical Pacific (i.e., the Southern Oscillation) and a quasi-periodic Sea Surface Temperature (SST) warming/cooling in the tropical eastern Pacific (i.e., El Niño/La Niña). The NAO is referred to as the meridional seesaw of the SLP field with the north and south centers near Iceland and the Azores, respectively. The PDO is characterized by a long-lived ENSO-like pattern that shifts phases with a period of at least 15 to 25 years.

The complete summary of the physical definition of all the 14 indices used in the study will be added to the revised article.

All of the indices have been proven to show certain signs of connections to East Asian climate variability in general (e.g., precipitation, monsoon and cyclone activity across seasons; see original manuscript P5L1–14), which is the main decision principle of the indices. Very likely only a few indices are relevant to Taiwan's hydro-climate variability (summertime streamflow in particular), and one way to figure out the significant indices is through our correlation analysis.

6) The authors clearly showed that Taiwan JAS catchment streamflow is significantly correlated with the WP and PJ teleconnection indices and the correlation relationship changes with time. The large-scale climate also shows decadal-scale variations. However, time coincidence cannot be used for arguing physical relationship. For example, the argument of "The CRS firstly emanates from the change in the basin-scale climatology over the Pacific (e.g., shift in the PDO), and then the reorganized large-scale patterns can reset the relationship between the island-scale streamflow with established regional circulations (e.g., the PJ pattern)" on P9L12-14 is a hand-waving argument. Nothing is explained about how the decadal-scale changes of sea surfaces temperature (PDO) influences the regional circulation pattern (PJ) in the atmosphere and then the rainfall pattern and subsequently the streamflow pattern in Taiwan. There are many more hand-waving type of argument in the paper. There is no point to list out all of them here if the above points are not addressed.

Whilst time coincidence cannot explain any physical relationship, it certainly motivates us to look into whether the physical relationship does exist. We wish to state that it is not our pure speculation by providing more explanation *and* evidence (references) below.

Since the PDO is strongly tied to ENSO, the shift in the PDO can induce changes in the ENSO-related SST anomalies as well as ENSO-related teleconnections (Duan et al., 2013). Such SST anomalies have been found robust in the western North Pacific during summer (Alexander et al., 2002), and numerical model experiments have verified the ENSO-forced PJ pattern (Kosaka et al., 2013). Kubota et al. (2016) further pointed out the ENSO-PJ relationship strengthened after 1980, and then weakened after 2000, likely due to the phase shift in the PDO. When the ENSO-PJ relationship is more pronounced, the systematic impacts of the PJ pattern on TC activity, rainfall, and subsequently streamflow in Taiwan are clear. By contrast, if the PJ pattern is less forced by ENSO, the associated impacts can be ambiguous. Consequently, Taiwan's streamflow become less predictable after the post-regime shift epoch.

We will supplement the above information to the revised article to make our argument sounder.

**References**

Alexander, M.A., Bladé, I., Newman, M., Lanzante, J.R., Lau, N.C. and Scott, J.D., 2002. The atmospheric bridge: The influence of ENSO teleconnections on air-sea interaction over the global oceans. Journal of Climate, 15(16), pp.2205-2231.

Chen, J.M. and Chen, H.S., 2011. Interdecadal variability of summer rainfall in Taiwan associated with tropical cyclones and monsoon. Journal of Climate, 24(22), pp.5786-5798.

Duan, W., Song, L., Li, Y. and Mao, J., 2013. Modulation of PDO on the predictability of the interannual variability of early summer rainfall over south China. Journal of Geophysical Research: Atmospheres, 118(23).

Hastenrath, S., 1995. Recent advances in tropical climate prediction. Journal of Climate, 8(6), pp.1519-1532.

Hastenrath, S., Polzin, D. and Camberlin, P., 2004. Exploring the predictability of the 'short rains' at the coast of East Africa. International journal of climatology, 24(11), pp.1333-1343.

Gerrity, J.P., 1992, A Note on Gandin and Murphy's Equitable Skill Score., Monthly Weather Review, 120, 2709-2712.

Wu, B., Zhou, T. and Li, T., 2009. Seasonally Evolving Dominant Interannual Variability Modes of East Asian Climate*. Journal of Climate, 22(11), pp.2992-3005.